# The physiological interactome of TCR-like antibody therapeutics in human tissues

Estelle Marrer-Berger[1], Annalisa Nicastri[2,3,8], Angelique Augustin[1,8], Vesna Kramar[4,8], Hanqing Liao[2,3], Lydia Jasmin Hanisch [4], Alejandro Carpy[5], Tina Weinzierl[4], Evelyne Durr[1], Nathalie Schaub[1], Ramona Nudischer[1], Daniela Ortiz-Franyuti[1], Ekaterina Breous-Nystrom[1], Janick Stucki[6], Nina Hobi[6], Giulia Raggi[6], Lauriane Cabon[1], Emmanuelle Lezan[1], Pablo Umaña [4], Isaac Woodhouse [2,3], Alexander Bujotzek [5], Christian Klein [4] ✉ & Nicola Ternette [2,3,7] ✉

Selective binding of TCR-like antibodies that target a single tumour-specific peptide antigen presented by human leukocyte antigens (HLA) is the absolute prerequisite for their therapeutic suitability and patient safety. To date, selectivity assessment has been limited to peptide library screening and predictive modeling. We developed an experimental platform to de novo identify interactomes of TCR-like antibodies directly in human tissues using mass spectrometry. As proof of concept, we confirm the target epitope of a MAGE-A4-specific TCR-like antibody. We further determine cross-reactive peptide sequences for ESK1, a TCR-like antibody with known off-target activity, in human liver tissue. We confirm off-target-induced T cell activation and ESK1-mediated liver spheroid killing. Off-target sequences feature an amino acid motif that allows a structural groove-coordination mimicking that of the target peptide, therefore allowing the interaction with the engager molecule. We conclude that our strategy offers an accurate, scalable route for evaluating the non-clinical safety profile of TCR-like antibody therapeutics prior to first-in-human clinical application.

T lymphocytes play a critical role in cancer immunity. Their ability to elicit antigen-mediated cytotoxicity has become a key therapeutic strategy to engage the immune system to fight cancer. Peptides presented by human leukocyte antigen (HLA) are the natural target of T cells and represent the largest pool of cell surface-presented cancer-specific targets. Therapeutically targeting specific peptide epitopes in complex with HLA complexes (HLAp) is thought to provide greater tumor selectivity over healthy tissues than targeting whole protein surface-expressed targets[1,2].

Bispecific antibodies that bind to a single HLA-bound peptide surface antigen on cancer cells and an activating T cell antigen such as CD3ε can, through the forced interaction, trigger the lysis of cancer cells. While a vast majority of the CD3 bispecific antibodies developed in cancer target surface oncoproteins, the recent FDA approval of

[1]Roche Pharma Research & Early Development, Roche Innovation Center Basel, 4070 Basel, Switzerland. [2]The Jenner Institute, Old Road Campus Research Building, Oxford OX37DQ, UK. [3]Centre for Immuno-Oncology, Old Road Campus Research Building, Oxford OX37DQ, UK. [4]Roche Innovation Center Zürich, 8952 Schlieren, Switzerland. [5]Roche Pharma Research & Early Development, Roche Innovation Center Munich, 82377 Penzberg, Germany. [6]Alveolix AG, Swiss Organs-on-Chip Innovation, 3010 Bern, Switzerland. [7]Department of Pharmaceutical Sciences, University of Utrecht, 3584 CH Utrecht, The Netherlands. [8]These authors contributed equally: Annalisa Nicastri, Angelique Augustin, Vesna Kramar. ✉e-mail: christian.klein.ck1@roche.com; nicola.ternette@ndm.ox.ac.uk

Tebentafusp, the first bispecific gp100 peptide-HLA-directed CD3 T cell engaged, provides the first clinical Proof of Concept of this approach for solid tumor therapy[3,4].

An inherent downside of TCR-based therapies and TCR-like antibodies is their potential to cross-react with HLA molecules presenting related peptides besides the desired one, which can result in severe off-target effects and organ toxicities[1,5,6]. Reported clinical fatalities related to off-epitope targeting with adoptive T cell therapies underpin the paramount importance of defining the unique specificity of these therapeutic molecules for their target before entering clinical trials. For TCR-like molecules, the strict human specificity of the HLA-I molecules, together with the limited overlap in protein processing and resulting peptide repertoires between humans and other species, invalidates the conventional toxicity testing in animals.

In the absence of a human relevant cross-reactive toxicology species, the non-clinical safety evaluation of TCR-like molecules is generally supported by new alternative methods. At an early stage of development, in silico strategies integrating peptide binding prediction and TCR/antibody contact profile elucidation through structural computational modeling are leveraged to predict potential off-targets[7]. In silico strategies interrogating TCR repertoires rely on interactome data obtained from large peptide or HLA peptide complex libraries that enable the determination of the TCR or TCR-like antibody specificity profile[8–15]. Further advances were made by bioinformatic approaches relying entirely on prediction of TCR interactomes based on TCR sequence features and their interacting peptide target[16–20]. However, it has been recognized that these approaches are still limited as the specificity of prediction is low, and hundreds of sequence candidates need to enter validation assays with generally low confirmed off-target hit rates. Furthermore, TCR specificity is expanded by a dramatic HLAp structural adaptability, demonstrated by unanticipated rearrangements in the peptide and presenting HLA protein[21]. These limitations and gaps highlight the need for novel, physiologically relevant technologies for off-target sequence identification[22].

Although human primary cells and organ models provide a unique opportunity to evaluate if the TCR-like engagers also trigger any relevant T cell effector function such as T cell-dependent lysis, it remains therefore, challenging to identify the causality of the cross-reactivity in primary healthy human tissues. In addition, most of the human model systems available still bear physiological and cellular gaps, raising the need to assess off-target binding in situ rather than in vitro.

Thus, to further increase the confidence in the selectivity of clinical TCR-like therapeutic candidates, we here adapt a mass-spectrometry technology-based approach to enable the systematic and robust identification of physiologically relevant off-target peptide-HLA complexes bound by TCR-like molecules in situ[23]. We utilize this technology to support the development of a TCR-like antibody targeting HLA-A*02:01 in complex with peptide GVYDGREHTV originating from MAGE-A4. Using this approach, we are further able to define causal off-targets of ESK1, a well characterized TCR-like T cell bispecific antibody, which is known to unselectively target peptide RMFPNAPYL derived from the intracellular oncoprotein WT1 (WT1[126-134]) presented on HLA-A*02:01[24–26], with high specificity. We demonstrate how a mimic motif engaging two separate positions in the peptide sequence allows off-target interaction with ESK1 and T cell activation which had been missed in previous prediction-validation approaches. Our data further demonstrate a high binding selectivity of the MAGE-A4 antibody.

Together, we show that our platform represents an additional strategy to assess cross-reactivities of TCR-like molecules, and that this approach may assist the development of highly specific and safe immunotherapies.

## Results

### A functional HLA-A2-specific MAGE-A4 epitope targeting TCR-like T cell bispecific (TCB) antibody

We generated antibodies specific for the MAGE-A4 peptide (GVYDGREHTV, MAGA4[230-239]) HLA-A*02:01 complex using phage display technology (further details are given in the Methods section). The 57D03 antibody was selected due to its selectivity profile and characterization in cell assays and reformatted into a trivalent 2 + 1 IgG TCB antibody with bivalent recognition of the MAGE-A4–peptide-MHC complex and monovalent binding to the CD3ε chain of the TCR. This was facilitated by fusion of a second fragment antigen-binding region (Fab) to the N-terminus of the variable-light (VL) domain of the anti-CD3 Fab, resulting in a heterodimeric 2 + 1 antibody (CrossMAbVH-VL)[27]. Introduction of P329G and L234A-L235A (PG LALA) mutations renders it inactive for immune effector functions[27–30]. For the work presented here, a CD3 binder based on the humanized antibody V9 derived from UCHT1, was used, yielding HLA-A2 MAGE-A4 targeting TCR-like TCB (henceforth referred to as MAGE-A4 TCB, Fig. 1A). Association and dissociation constants for HLA-A*02- GVYDGREHTV and CD3ε are summarized in Supplementary Fig. 1.

Our aim was then to confirm the functionality of the MAGE-A4 TCB in HLA-A*02:01-positive malignant melanoma cell line A375 which expresses MAGE-A4. We decided to optimize our assay using A375 cells and xenograft material to demonstrate killing both in tumor cells and tissue material. We first confirmed and quantified presentation of the MAGE-A4 target peptide sequence by HLA-A*02:01 on A375 cells using a standard, targeted immunopeptidomics approach and determined the target peptide abundance to a minimum of 30-50 copies per cell in cells and xenografts (Fig. 1B). The experiment was controlled by a parallel investigation using A375 xenografts in which the target antigen, MAGE-A4, had been removed by CRISPR-KO (A375-KO). We finally confirmed that the measured interaction of the MAGE-A4 TCB with the target peptide induces the killing of the A375 cells when co-cultured with human PBMCs. We determined concentration-dependent, highly effective lysis of A375 cells, validating the functionality of the TCB (Fig. 1C).

### De novo identification of on-target binding of the MAGE-A4 TCR-like antibody to its target peptide GVYDGREHTV using an experimental framework

To demonstrate the on-target binding of the MAGE-A4 antibody to the target HLAp, we developed a immunopeptidomics strategy that uses the HLAp-targeting antibody as a bait to enrich interacting HLAp from A375 xenografts (Fig. 1D). For this purpose, we immobilized the IgG-format of the MAGE-A4 TCR-like antibody (henceforth referred to as MAGE-A4 antibody) on Protein A beads by cross-linking, and exposed the molecule to the solubilized proteome, including membrane-bound HLAp, of the A375 xenografts. We subsequently eluted the interacting HLAp bound to the MAGE-A4 antibody and purified the HLA-associated peptides through a 5 kDa molecular weight filter before liquid chromatography-tandem mass spectrometry (LC-MS[2]) analysis for sequence determination (Fig. 1D). The experiment was controlled by an isotype control antibody targeting an irrelevant HLAp, and A375-KO xenograft tissue as before. The target peptide of the MAGE-A4 antibody, GVYDGREHTV, was detected in all three replicate analyses in A375 xenografts with high confidence, and was not identified in the A375-KO nor when an isotype control was used as a bait, as expected (Fig. 1E, F). It is noteworthy that in addition to the target peptide, a shorter version of the target peptide (YDGREHTV) was detected with lower intensity (Supplementary Fig. 2).

Out of all detected peptides predicted to bind to HLA-A*02:01, GVYDGREHTV was the only sequence that was identified as significantly enriched by the MAGE-A4 antibody in A375 xenografts, indicating a highly selective binding capacity of the antibody (Supplementary Data 1). These results show that we were able to enrich and identify HLA-peptide complexes using TCBs in IgG format, and that we can de novo identify the target peptide sequence by LC-MS[2] using this approach.

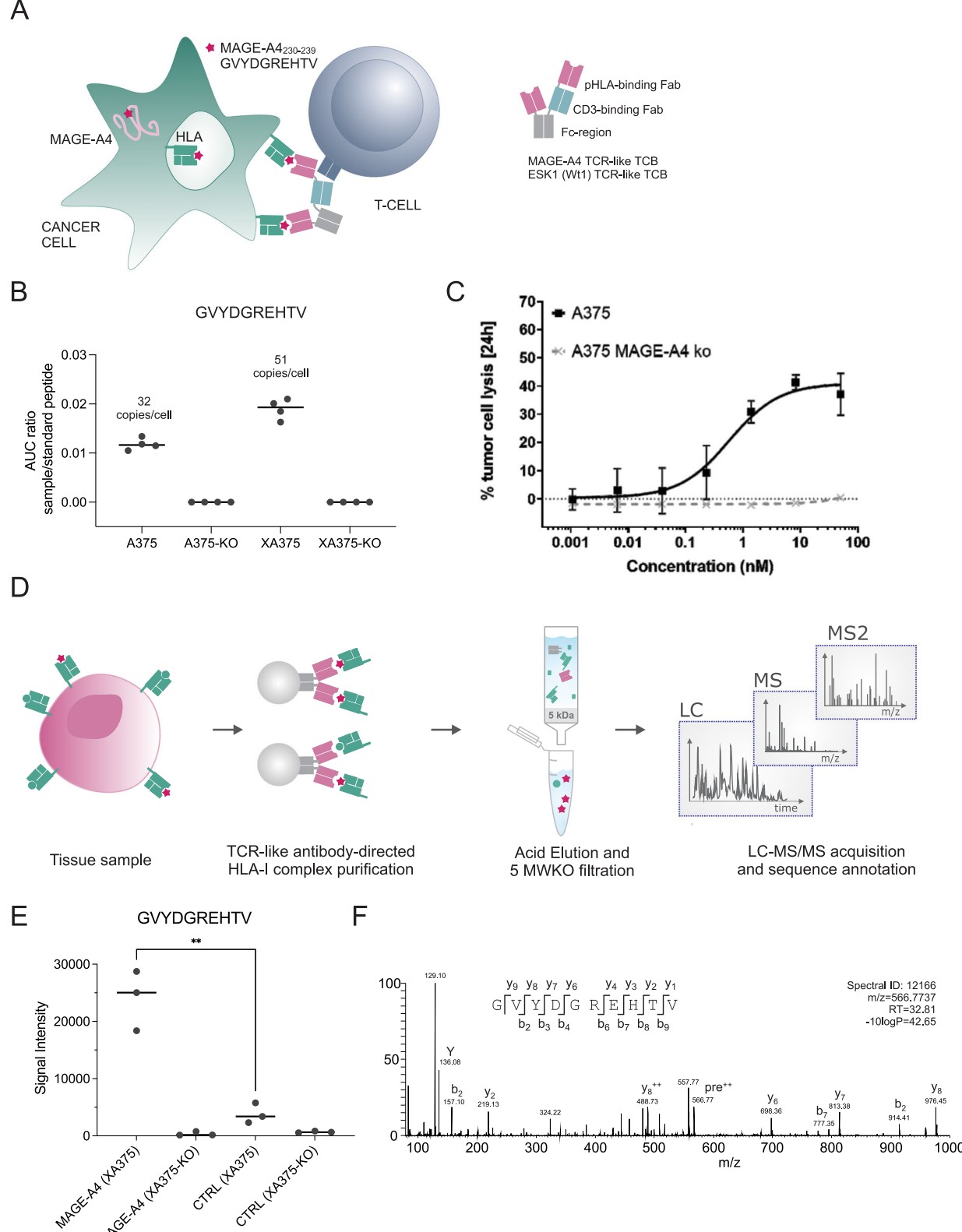

## Direct comparison of interactomes of MAGE-A4 to ESK-1, a known TCR-like antibody with off-target activity identifies ESK1 cross-reactive peptides in healthy liver tissue

We then aimed to expand our assay from xenograft tissue to healthy human tissues for the evaluation of off-target reactivity. Next to our clinical candidate, we decided to include a second TCR-like antibody, ESK1, as a control. ESK1 was developed to recognize HLA-A*02:01 presented peptide derived from the intracellular oncoprotein WT-1 (RMFPNAPYL, WT1$_{126-134}$)[24–26]. The antibody has never entered clinical development due to off-target binding and lysis of diverse human primary cells. Candidate off-target sequences have been proposed recently[25,26], however, the presentation of causal off-target peptide sequences in primary human tissues has not been confirmed to date.

**Fig. 1 | An experimental framework de novo identifies on-target binding of the functional MAGE-A4-specific TCB recognizing the peptide GVYDGREHTV presented by HLA-A*02:01. A** Schematic of the MAGE-A4 TCB and its mode of action. **B** Average ratio of endogenous GVYDGREHTV peptide area under the curve (AUC) by comparison to spiked-in isotopically labeled GVYDGREHTV peptide AUC (n = 4), measured in elution fractions from HLA-A2 immunoprecipitation in A375 tumor cells lines and MAGE-A4 KO (A375-KO) and equivalent lines in a xenograft model (XA375 and XA375-KO). Horizontal bars depict the mean of quadruplicate analyses. The estimated copy number of endogenous GVYDGREHTV per cell (shown on top of each bar graph) is calculated by comparison to a spiked-in isotopically labeled GVYDGREHTV peptide, assuming 100% recovery for both standard and endogenous peptides at the end of the workflow. **C** A375 wild-type cells and A375-KO cells were incubated with human PBMCs at an E:T ratio of 10:1. Depicted are dose-response curves of tumor cell lysis after 24 h of incubation with different concentrations of MAGE-A4 TCB as determined by quantification of lactate dehydrogenase release in the supernatant (plotted are the mean of triplicate repeat

analyses and error bars depict the SD). **D** Schematic workflow using TCR-like antibodies to enrich interacting HLA-peptide complexes. Tissues were lysed, and solubilized HLA-peptide complexes were immunoprecipitated using the MAGE-A4 tool antibody as a bait. Peptides were enriched in acidic conditions through a 5 kDa molecular weight cut-off (MWKO) filter and analyzed by LC-MS[2]. **E** Peptide intensity as measured in three biological replicate analyses demonstrates significant enrichment (p = 0.0015, unpaired t-test, two-tailed) of the target peptide GVYDGREHTV in XA375, and not in MAGE-KO equivalent tissues (XA375-KO) nor with a control TCB (CTRL) (**F**) LC-MS spectrum leading to identification of the target peptide GVYDGREHTV in XA375 after enrichment using the MAGE-A4 antibody. The spectral identifier, the measured mass of the precursor peptide ion (m/z), charge state (z), and retention time (RT), at which the peptide ion was selected for fragmentation, are stated within the spectral panel. C-terminal fragment ions are indicated as y, N-terminal fragments are designated b, pre: unfragmented precursor peptide, +: singly charged ion, ++: doubly charged ion.

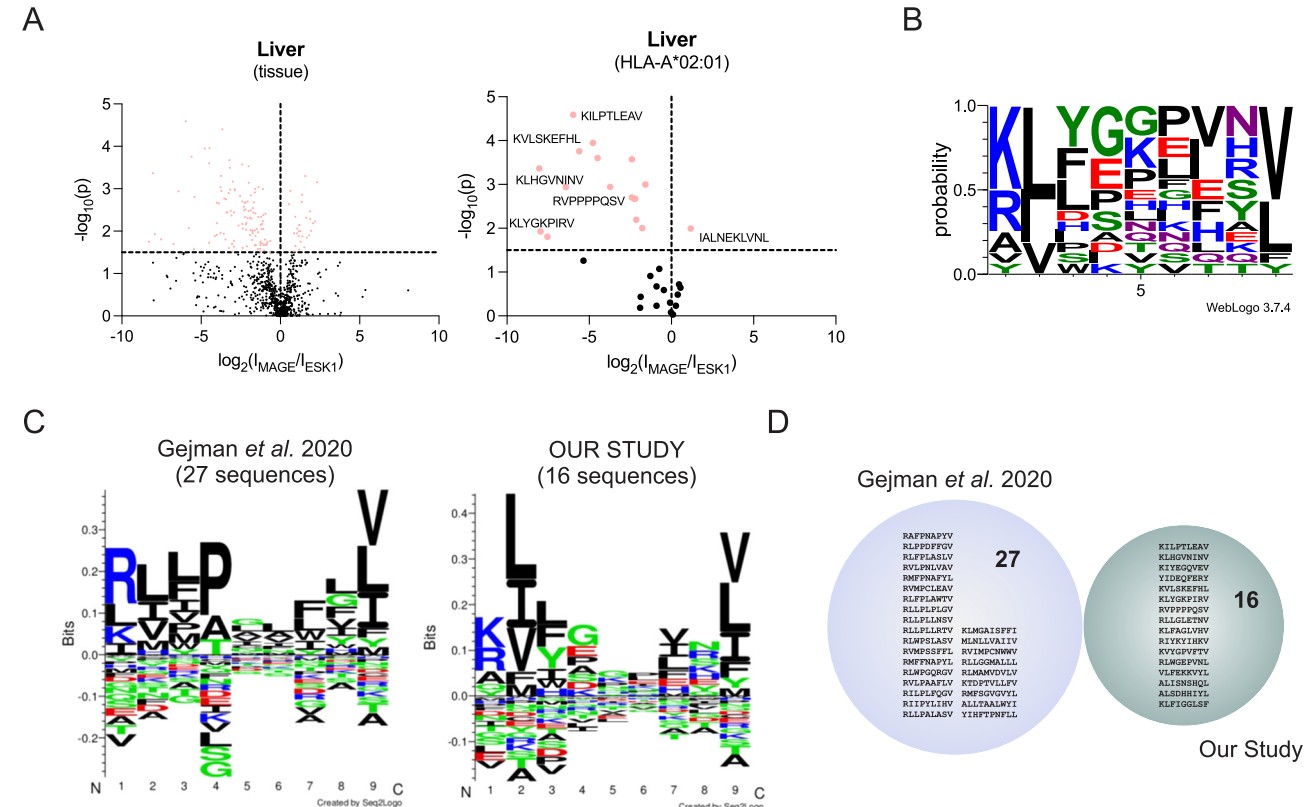

**Fig. 2 | Off-target interaction of ESK1 with common HLAp sequences in primary liver tissue. A** Volcano plots showing significant (one-factor ANOVA, -log10(p)≥1.5) enrichment of peptides for ESK1 and MAGE-A4 in liver tissue (all peptides, left panel; peptides predicted to bind to HLA-A*02:01, right panel). **B** Sequence motif of the ESK1-enriched peptides that are predicted to bind to HLA-A*02:01. Shown is the

frequency (probability) of each amino acid in each position across the peptide sequence from N- to C-terminus (left to right, positions 1-9). **C** Sequence motifs from Gejman et al. (left panel) and our study (right panel). **D** Venn diagram depicting the lack of overlap of peptide sequences shortlisted by Gejman et al. and our study as Esk1 binders.

We chose to perform a head-to-head immune-enrichment from primary liver tissue resections using both, MAGE-A4 and ESK1 antibodies. Since both target antigens MAGE-A4 and WT-1 are not expressed in hepatocytes (Human Protein Atlas, www.proteinatlas.org), the ESK1 enrichment could serve as isotype control for the MAGE-A4 antibody, and vice versa. The comparison of significantly enriched interactomes for both antibodies uncovered 16 HLA-A*02:01 peptides for Esk1, and 1 peptide for the MAGE-A4 antibody (Fig. 2A, Table 1, Supplementary Data 2). The sequence motif of peptides significantly enriched by ESK1 shows a preference for a R/K/A in the first position, while A2 anchor residues are maintained (Fig. 2B). This observation is in alignment of the previous observation that ESK1 interacts mainly with the N-terminal domain of the HLA-A*02:01-bound peptide[26].

We then proceeded to compare our identified HLAp sequences with off-target sequences published by Gejman et al. [25]. In this study, a genetic minigene platform (PresentER) was used to experimentally test predicted peptide sequences for interaction with ESK1. The authors identified 100 peptide sequences to be interacting with ESK1, of which 27 entered validation assays. We observed similarities in the obtained weighted sequence motif of the 27 shortlisted peptide sequences from these experiments and the weighted motif determined in our study (Fig. 2C). Both motifs accurately reflect the first four amino acids of the ESK1 target peptide sequence RMFPNAPYL (R-M-F-P are present in the obtained motif at positions 1-4, Fig. 2C), which is not present in healthy liver tissue. Nine out of the 27 Gejman et al. [25] reported peptides are known to be expressed in liver (Supplementary Table 1), however,

**Table 1 | TCB-enriched peptides identified in primary liver tissue**

| Name | Sequence | A*02:01 Rank | Accession | Protein name | Intensity ratio (ESK1/MAGE) | -log$_{10}$(p) | Esk1 JNFAT signal |
|---|---|---|---|---|---|---|---|
| ILF2$_{127-135}$ | **KILPTLEAV** | 0.051 | Q12905 | Interleukin enhancer-binding factor 2 | 64.0 | 4.59 | + |
| RBM4B$_{59-67}$ | **KLHGVNINV** | 0.03 | Q9BQ04 | RNA-binding protein 4B | 84.4 | 2.94 | ++ |
| RL5$_{117-125}$ | KIYEGQVEV | 0.004 | P46777 | 60 S ribosomal protein L5 | 4.6 | 2.67 | - |
| SEPT2$_{121-129}$ | YIDEQFERY | 1.77 | Q15019 | Septin-2 | 4.3 | 2.19 | - |
| SET$_{150-158}$ | KVLSKEFHL | 0.324 | Q01105 | Protein SET | 48.5 | 3.76 | - |
| SF3B4$_{78-86}$ | **KLYGKPIRV** | 0.004 | Q15427 | Splicing factor 3B subunit 4 | 238.9 | 1.93 | + |
| SHC1$_{469-477}$ | **RVPPPPQSV** | 0.183 | P29353 | SHC-transforming protein 1 | 5.3 | 2.7 | ++ |
| ARHGEF26$_{863-871}$ | **RLLGLETNV** | 0.059 | Q96DR7 | Rho guanine nucleotide exchange factor 26 | 181.0 | 1.8 | ++ |
| CRYL1$_{130-138}$ | **KLFAGLVHV** | 0.004 | Q9Y2S2 | Lambda-crystallin homolog | 256.0 | 3.36 | ++ |
| MSMO1$_{164-172}$ | RIYKYIHKV | 0.03 | Q15800 | Methylsterol monooxygenase 1 | 13.0 | 2.94 | - |
| CYP2C8$_{59-67}$ | **KVYGPVFTV** | 0.011 | P10632 | Cytochrome P450 2C8 | 22.6 | 3.6 | ++ |
| USP9Y$_{1674-1682}$ | **RLWGEPVNL** | 0.014 | O00507 | Prob. ubiquitin carboxyl-terminal hydrolase FAF-Y | 27.9 | 3.95 | ++ |
| PLG$_{92-100}$ | VLFEKKVYL | 0.008 | P00747 | Plasminogen | 4.9 | 2.67 | - |
| MAF$_{116-124}$ | ALISNSHQL | 0.038 | O75444 | Transcription factor Maf | 3.0 | 3 | - |
| ALDOA$_{216-224}$ | ALSDHHIYL | 0.011 | P04075 | Fructose-bisphosphate aldolase A | 3.5 | 2 | - |
| HNRNPA1$_{15-23}$ | KLFIGGLSF | 1.625 | P09651 | Heterogeneous nuclear ribonucleoprotein A1 | 5.3 | 3.57 | - |
| EIF3F$_{348-357}$ | IALNEKLVNL | 0.613 | O00303 | Eukaryotic translation initiation factor 3 subunit F | 0.4 | 1.99 | - (MAGE-A4 TCB) |

+: Positive signal in JNFAT assay, ++: Positive signal in JNFAT assay higher than ESK1-target peptide, -: negative in JNFAT assay. Bold peptide sequence font: Positive signal in the presence of ESK1-TCR in JNFAT assay. -log$_{10}$(p): designates negative decadic logarithm of the one-factor ANOVA p value (assuming unequal variances).

none of the sequences were confirmed in our assays as physiological ligands presented by HLA in healthy (non-cancerous) human liver tissue (Fig. 2D).

**Half of the identified ESK1-cross reactive peptides are potent to elicit T cell activation**

The 16 peptides which were enriched by the ESK1 antibody were further evaluated for their potential recognition by ESK1 TCB using Jurkat NFAT activation assay. T2 cells were first pulsed with the peptides and then co-incubated with JNFAT effector cells in the presence of different concentrations of ESK1 TCB. The target peptide RMFPNAPYL (WT1$_{126-134}$) was used as a positive control, while VLDFAPPGA, a WT1-derived non-relevant peptide (WT1$_{37-45}$) was used as negative control. Eight peptides resulted in a positive signal in two independent experiments. While the known targeted RMFPNAPYL peptide elicited the expected signal, SHC1$_{469-477}$, RMB4B$_{59-67}$, USP9Y$_{1674-1682}$, ARHGEF26$_{863-871}$ CRYL1$_{130-138}$, and CYP2C8$_{59-67}$, gave a higher signal than the RMFPNAPYL target peptide sequence in the JNFAT reporter assay, while ILF2$_{127-135}$ and SF3B4$_{78-86}$ gave a positive signal that was weaker than that of the target peptide in direct comparison (Fig. 3A). Furthermore, the signal of the 6 peptides that showed higher T cell activation in the presence of ESK1 TCB remained consistently higher than those of the RMFPNAPYL target peptide even at 100-fold diluted peptide concentrations (Fig. 3B). Interestingly, the ESK1-reactive peptides all exhibited a high fold-enrichment in the interactome assay (≥ 4.5-fold enrichment) with one exception, SHC1$_{469-477}$, which was 2.4-fold enriched but conversely resulted in the highest activation response. We did not observe a significant correlation between the measured fold enrichment and the degree of T cell activation (Supplementary Fig. 3). All peptides were also evaluated for their potential recognition by MAGE-A4 TCB using the same assay, including peptide EIF3F$_{348-357}$ which had been detected as enriched in by the MAGE-A4 TCB in the interactome assay. The target peptide (MAGA4$_{230-239}$) was used as a positive control, while WT1$_{37-45}$ and WT1$_{126-134}$ were used as negative controls. No signal was observed with MAGA4 TCB other than with its target peptide

sequence (Fig. 3C). The peptide which was most enriched in the MAGE-A4 antibody fraction that was not predicted to bind to HLA-A*02:01 was also tested for its potential to activate NFAT Jurkat cells, and, as expected, no signal was observed (Supplementary Fig. 4).

**ESK1 off-target activity leads to killing of liver spheroids**

To confirm that the observed cross-reactivities of ESK1 TCB in situ are relevant for killing of primary hepatocytes, and to validate the high specificity observed for the MAGE-A4 antibody, we employed an in vitro T cell dependent lysis assay performed with a 3D HLA-A*02:01 human liver spheroid cell model. The in vitro killing assays were performed by exposing co-cultures of liver spheroids and primary allogenic PBMCs to the relevant TCBs. The assays were controlled by testing co-cultures upon incubation in medium only and assaying the negative control DP47 TCB (a non-tumor targeted TCB) side by side with MAGE-A4 TCB and ESK1 TCB. The absence of expression of both target proteins, MAGE-A4 and WT1, in the liver spheroids allowed the exclusion of a target-mediated effect by MAGE-A4- and ESK1 TCBs, respectively. Cytotoxicity was assessed by live imaging using measurement of Granzyme B secretion, and by monitoring the levels of aspartate aminotransferase. As expected by the in situ cross-reactivity data, ESK1 TCB triggered significant dose-dependent target cell lysis accompanied by T cell activation/proliferation and cytokine release in liver spheroid tissue (Fig. 3D). In alignment with our observation that the MAGE-A4 TCB has high selectivity for its target peptide sequence, no significant effects were observed in this model, suggesting the absence of off-target cross-reactivity.

**ESK1 off-target sequences feature a compensation binding motif, which results in a similar structural conformation than the target peptide and allows interaction with the engager molecule despite complete amino acid sequence divergence**

To explore how ESK1 might be able to bind to the off-target peptides when being presented on HLA-A*02:01, we generated a molecular model of all of the Jurkat-NFAT confirmed eight peptides in complex

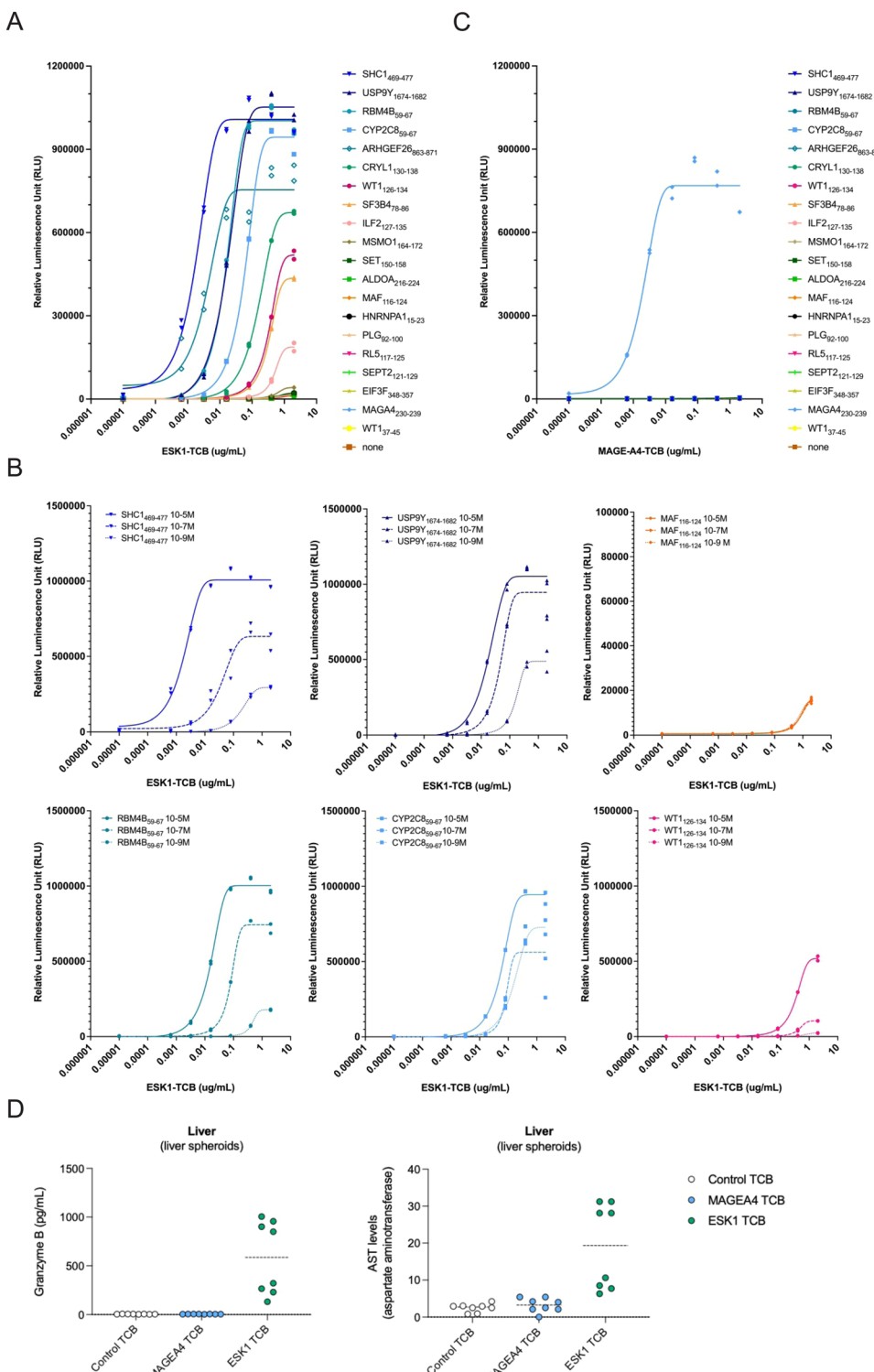

**Fig. 3 | Off-target binding activates effector cells and conveys killing of liver spheroids in the presence of ESK1.** **A** T2 cells were first pulsed with the 16 ESK1-enriched peptides and then co-incubated with JNFAT effector cells in the presence of different concentrations of ESK1 TCB. The target peptide (WT1$_{126-134}$) was used as a positive control, while WT1$_{37-45}$, a non-relevant peptide, was used as a negative control. Effector cell activation was evaluated after 6 h by measuring luminescence signal using ONEGlo. **B** Titration curves for JNFAT assay with indicated peptides and

ESK1 TCB concentrations. **C** Equivalent to (A) using MAGE-A4 TCB, and MAGE-A4 target peptide MAGA4$_{230-239}$ as a control. **D** Liver spheroids were cocultured with peripheral blood mononuclear cells at a ratio of 5 effector to 1 target cell and target cell killing was evaluated by granzyme B secretion and aspartate aminotransferase expression. Each graph represents two independent experiments with two human donors in quadruplicates.

with HLA-A*02:01 and in the presence of the Fab of ESK1, based on the X-ray crystal structure (PDB-ID 4WUU) of the ESK1 Fab binding to HLA-A*02:01-WT1$_{126-134}$ (RMFPNAPYL). These analyses show that all cross-reactive peptides can be accommodated in the structure without

significant steric clashes (Fig. 4A). In the structural alignment, the four N-terminal peptide positions are showing a high degree of similarity with regards to backbone conformation and sidechain orientation, while the central part of the peptide (positions 5-7) is structurally

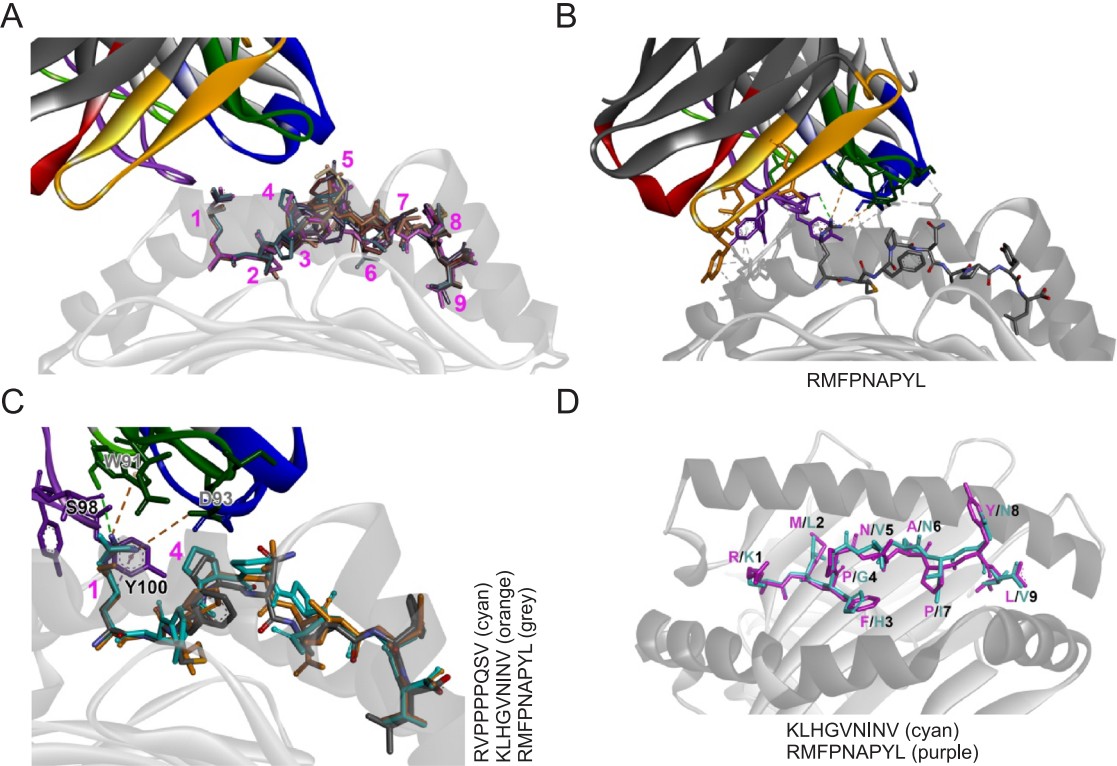

**Fig. 4 | Structural mimicry is a feature of ESK1 off-target activity. A** Overlay of RMFPNAPYL (purple, from X-ray structure with PDB ID 4WUU) with models of the confirmed cross-reactive peptides in the presence of the Fab of ESK1. The HLA-A*02:01 peptide binding groove is indicated in gray. The numbers indicate the approximate position of the amino acid side chain at the respective peptide position. **B** RMFPNAPYL peptide within the HLA-A*02:01 binding groove and interactions with the ESK1 Fab (PDB ID 4WUU). **C** Non-bonded interactions between the Fab of ESK1 and RMFPNAPYL (gray) in the HLA-A*02:01 peptide-HLA complex (X-ray crystal structure with PDB ID 4WUU). Models of the cross-reactive peptides SHC1$_{469-477}$ (cyan) and RBM4B$_{59-67}$ (orange) are shown for comparison. **D** Overlay of RMFPNAPYL peptide (purple, from X-ray structure with PDB ID 4WUU) with modeled RBM4B$_{59-67}$ peptide (cyan) in the peptide binding groove of HLA-A*02:01.

divergent. The amino acid on position 8, as well as the anchor residue on position 9, are again tightly aligned.

The crystal structure of ESK1 binding to the HLA-A*02:01-WT1$_{126-134}$ complex documents that ESK1 is binding in the proximity of the N-terminal half of the RMFPNAPYL peptide (Table 2), and that direct chemical interactions between ESK1 and the RMFPNAPYL peptide are only occurring with the arginine residue on position 1 (Fig. 4B). Further interactions between position 4 of the peptide and CDR-L3 residue D93 of ESK1 are conceivable due to their short distance of only 3.4 Å.

The most active peptide SHC1$_{469-477}$ (RVPPPPQSV) is identical to WT1$_{126-134}$ (RMFPNAPYL) on the critical positions 1 and 4. This provides a clear structure-based rationale for the recognition by ESK1 via the RXXP motif and the observed binding mode (Fig. 2C). However, and in marked contrast to SHC1$_{469-477}$, three out of the eight peptides did not share a single amino acid at any given position with the WT1$_{126-134}$ target peptide (RMB4B$_{59-67}$ (KLHGVNINV), CYP2C8$_{59-67}$ (KVYGPVFTV), SF3B4$_{78-86}$ (KLYGKPIRV), Table 2). The most reactive peptide amongst these, RBM4B$_{59-67}$ (KLHGVNINV), was able to adapt a very similar conformation in the peptide binding groove (Fig. 4C, D). Residue G4 of the RMB4B$_{59-67}$ (KLHGVNINV) peptide is able to substitute for P at position 4 of the WT1$_{126-134}$ (RMFPNAPYL) peptide and promotes a turn in the backbone. Similarity is also found in position 1 (with R and K both carrying a positive charge) and position 3 (with F and H both being of similar size and having an aromatic ring). In addition, the same holds true for the pairs of hydrophobic amino acids found on the classical anchor positions 2 and 9. The same amino acid motif, KXXG, and an aromatic amino acid (H, Y) on position 3 [KXX$_{H/Y}$G], is also found for the two other cross-reactive peptides CYP2C8$_{59-67}$ (KVYGPVFTV) and SF3B4$_{78-86}$ (KLYGKPIRV) and consequently a very similar binding mode to RBM4B$_{59-67}$ can be expected.

Interestingly, the cross-reactive peptides ARHGEF26$_{863-871}$ (RLLGLETNV) and USP9Y$_{1674-1682}$ (RLWGEPVNL) seem to contain a hybrid of the RXXP and KXXG motifs observed here [RXX$_{L/W}$G]. Once more, G is confirmed as a favorable substitution for P on position 4, although the set of cross-reactive peptides also contains one instance of A on position 4 (CRYL1$_{130-138}$, KLFAGLVHV). This specificity profile of position 4 is in line with earlier binding studies of ESK1 involving mutated versions of the WT1$_{126-134}$ peptide[25].

Finally, the peptides shown to be non-activating in the JNFAT assay (Table 1) also share structural features that set them apart from the activating peptides: In all but one of the cases, this is a charged residue (3 x E, 1 x D, 1 x K) or a polar, uncharged residue (2 x S) at position 4. As an exemption to this rule, the non-activating peptide HNRNPA1$_{15-23}$ (KLFIGGLSF) has an I on position 4, which is hydrophobic (same as the A which is tolerated on this position) but seems to cause steric hindrance and non-favorable-,-chemical interactions with CDR-L3 of ESK1. Notably, the presence of these non-tolerated amino acids on position 4 seems to overrule the presence of a favorable residue on position 1 (3 x K, 1 x R).

Importantly, such dual-position-dependent amino acid motifs are not discoverable with conventional approaches like alanine-screening, and hence fundamentally extend the current experimental capabilities for TCR-like antibody specificity assessment.

## No off-target interactions detected between the MAGE-A4 TCB and lung and colon tissue

Finally, in order to extend the functionality of our approach for de-risking of the MAGE-A4 TCB, we applied our interactome assay to two further healthy tissues from lung and colonic tissue. We used an iso-type control targeting HLA-DQ, a low-expressed HLA class II surface

**Table 2 | Change of the solvent-accessible surface area (ΔSASA) of the RMFPNAPYL peptide in the absence of the ESK1 Fab (calculated from the X-ray crystal structure with PDB ID 4WUU) and comparison to the confirmed cross-reactive peptides, ordered by their signal in the JNFAT activation assay**

| Position | 1 | 2 | 3 | 4 | 5 | 6 | 7 | 8 | 9 |
|---|---|---|---|---|---|---|---|---|---|
| WT1$_{126-134}$ | **R** | **M** | **F** | **P** | **N** | **A** | **P** | **Y** | **L** |
| ΔSASA without ESK1 | 69.1 | 0.0 | 0.0 | 35.7 | 15.1 | 0.0 | 0.0 | 0.0 | 0.0 |
| Interacts with ESK1 | ✓ | ✗ | ✗ | (✓) | ✗ | ✗ | ✗ | ✗ | ✗ |
| SHC1$_{469-477}$ | **R** | V | P | **P** | P | P | Q | S | V |
| ARHGEF26$_{863-871}$ | **R** | L | L | G | L | E | T | N | V |
| RBM4B$_{59-67}$ | K | L | H | G | V | N | I | N | V |
| USP9Y$_{1674-1682}$ | **R** | L | W | G | E | P | V | N | **L** |
| CYP2C8$_{59-67}$ | K | V | Y | G | P | V | F | T | V |
| CRYL1$_{130-138}$ | K | L | **F** | A | G | L | V | H | V |
| SF3B4$_{78-86}$ | K | L | Y | G | K | P | I | R | V |
| ILF2$_{127-135}$ | K | I | L | **P** | T | L | E | A | V |

Bold font: Amino acid identical to respective residue in RMFPNAPYL peptide.

molecule. Only two peptides were significantly enriched by the MAGE-A4 antibody in lung tissue in these experiments (Fig. 5A, Supplementary Data 3, 4). Both peptides did not show the potential to activate Jurkat cells in the NFAT reporter assay, as well as the most enriched peptide which was not predicted to bind to HLA-A*02:01 (Supplementary Fig. 5). As expected, we also did not identify the target peptide, nor any other peptide from the source protein MAGE-A4, in any of the two healthy tissues interrogated here.

To confirm that the MAGE-A4 TCB does not activate T cells in primary human PBMC in the context of these tissues, we again performed in vitro killing assays. As before, we controlled the in vitro assays by testing MAGE-A4 TCB and ESK1 TCB side by side with a negative control DP47 TCB (a non-tumor-targeted TCB). MAGE-A4 and WT1 were not expressed in either lung or colonic 3D models. Cytotoxicity was assessed by live imaging using a Caspase 3/7 fluorescent probe, or a Granzyme B release assay as before. No significant off-target effects were detected for the MAGE-A4 TCB, mirroring the lack of specific peptide enrichment in either tissue observed with the MAGE-A4 antibody in the interactome assay (Fig. 5B). In contrast, ESK1 TCB again showed off-target activity in both lung and colonic 3D models (Fig. 5B).

We conclude that MAGE-A4 TCB is a therapeutic engager molecule with no observed off-target activity in liver, lung, and colon tissues.

## Discussion

An urgent priority for the development of HLA-peptide-specific immunotherapies is the establishment of strategies for off-target screening to de-risk these therapies in the absence of suitable in vivo strategies prior to entering human clinical trials. Animal models have limited validity due to species specific MHC binding specificities and an altered antigen processing machinery[31]. The definition of off-target peptide sequences by integration of library data, prediction and structural modeling has advanced our understanding of TCR interactomes but has not been sufficient to define physiologically relevant risk antigens in the past[22].

We present here an approach that is able to test the TCR-like molecule against the whole naturally processed and presented peptidome repertoire of a given primary human tissue, for which thousands of HLA-associated peptides can be identified by immunopeptidomics approaches. We use a TCR-like antibody enrichment strategy and an

LC-MS readout to evaluate the interactome of a TCR-like TCB targeting MAGA4$_{230-239}$ in the HLA-A*02:01 context. We confirmed the functionality of the approach in a proof-of-concept experiment, in which we demonstrated the significant enrichment of the target peptide using the TCR-like antibody as a bait in A375 xenograft tissue. In addition to the target peptide we also identified a shorter 9mer version of the target peptide at low intensity, which could result from MAGE-A4 antibody binding and co-enrichment but is likely a result from in-source fragmentation during MS acquisition[32].

We further investigate interactomes of the MAGE-A4 antibody and ESK1, a TCR-like TCB targeting WT1$_{126-134}$ which has not entered clinical development due to off-target binding and lysis of diverse human primary cells. Epitope binding of ESK1 relies almost exclusively on peptide residue Arg1[25,26]. Candidate off-target sequences have been proposed recently[25], however, the physiological presentation of off-target peptide sequences in primary human tissues has not been confirmed to date. Strikingly, using our methodology, we identify here 16 ESK1 interacting HLA-presented peptide sequences in primary healthy liver tissue - all confirmed HLA-presented and, therefore, physiologically relevant. Neither of these peptide candidates had been previously identified as an off-target candidate for ESK1, despite the fact that amino acid motifs for off-target sequences identified between our study and Gejman et al. [25] show a remarkable overlap (Fig. 2B), and that predicted sequence requirements for ESK1 recognition by Ataie et al. 2016 were in alignment with our observations. The most apparent difference between our study and Gejman et al. [25] was a strong requirement for a proline residue at position 4 of the peptide enabling interaction with ESK1, which could not be observed in our data. Instead, we determined a stronger requirement for an aromatic or hydrophobic amino acid in position three, despite a lack of evidence for this residue to be important for ESK1 Fab engagement.

We further observed the ability to induce T cell activation for eight out of the 16 (50%) identified peptide sequences in the presence of the TCR-like TCB in a Jurkat NFAT activation assay. Such exceptionally high specificity has not been achieved by any other methodology. Six of the active peptides induced higher T cell activation than the target peptide. These six peptides belong to the most significantly enriched peptides in the interactome assay, with high enrichment ratios. In depth analysis of the structure of the highest reactive peptide, SHC1$_{469-477}$ (RVPPPPQSV), did not immediately conclude why the peptide is exhibits higher activation profiles in situ than the RMFPNAPYL target sequence. Possibly, the quadruple P motif leads to a more favorable backbone conformation for recognition by ESK1 as P4 in the modeled complex is situated above than P4 in the RMFPNAPYL complex, and thereby more available for interactions with the CDR-L3 loop of ESK1 (Fig. 4C).

Three peptides, RMB4B$_{59-67}$ (KLHGVNINV), CYP2C8$_{59-67}$ (KVYGPVFTV), SF3B4$_{78-86}$ (KLYGKPIRV) did not have any sequence identity with the target peptide sequence and had a KXXG motif rather than the alternative RXXP recognition motif of the ESK1 target sequence. Birnbaum et al. showed in 2014 that unrelated peptide sequences can emerge from sequence clustering and structural analysis and conclude that TCR interactomes share many common features with regards to docking geometry and interaction chemistry[9]. The authors describe hot spots on a peptide, while tolerating extensive diversity at other positions. Dash et al. later confirm these observations by identification of shared core sequence similarities for a clustered group of receptors, but also observe a diverse set of outlier sequences[33]. It is, therefore, generally accepted that TCR cross-reactivity is shaped by structural adaptability of the HLA peptide complex rather than by sequence alone[21].

Neither of the remaining ESK1-enriched peptide sequences that were unable to elicit a JNFAT signal had either of the two detected sequence motifs RXX(P/G), KXX(G/P/A), suggesting that these motifs are in fact predictive for T cell activation via the ESK1 TCB. These

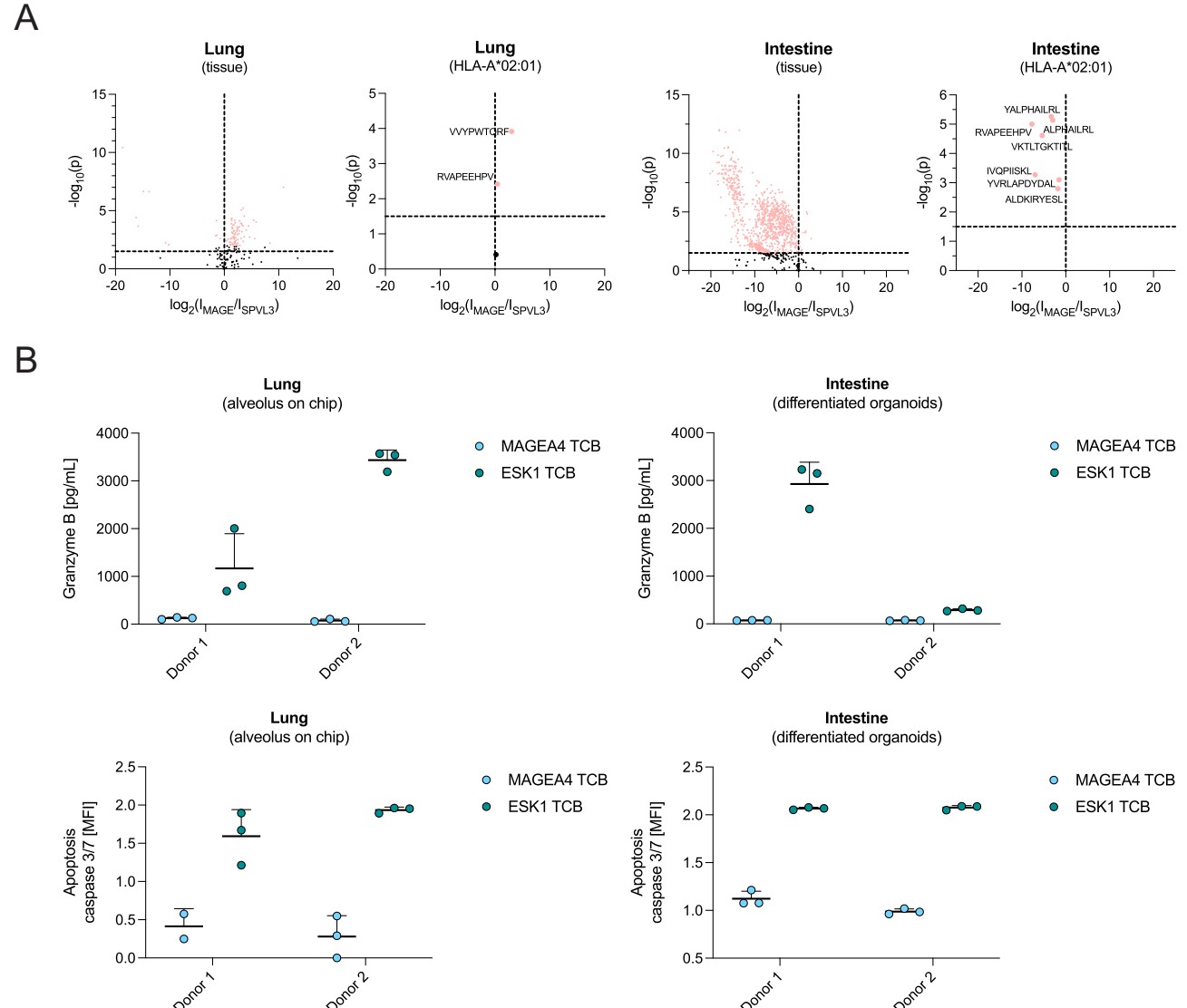

**Fig. 5 | MAGE-A4 TCB does not show off-target reactivity in lung and intestinal tissue. A** Volcano plots showing significant enrichment (one-factor ANOVA, $-\log_{10}(p) \geq 1.5$) of peptides for anti-HLA-DQ (clone SPVL3), upper left quadrants) and MAGE-A4 (upper right quadrants) in lung and intestine tissue (all peptide, left panels; peptides predicted to bind to HLA-A*02:01, right panels). **B** Lung alveolus-

on-chip and intestinal differentiated organoids were cocultured with peripheral blood mononuclear cells at a ratio of 10 effector to 1 target cell. Target cell killing was evaluated by granzyme B secretion and caspase 3/7 probe imaging (MFI median fluorescence intensity). Each graph represents two independent experiments with two human donors in triplicates.

results emphasize that the incorporation of the structural interphase of the HLA-peptide complex and TCR is essential to enhance predictive algorithms and highlight that reliance on sequence homology prediction is limited to capturing only a part of the potential interactome of a given TCR-like molecule.

We finally confirm that the observed cross-reactivities lead to killing in 3D healthy human liver cells in co-culture with ESK1 and PBMC, confirming the validity of our assay and the causality of the detected cross-reactive HLAp sequences. We further demonstrate that the high specificity observed for the MAGE-A4 antibody is accurate, as no significant killing in lung, colonic and liver cells is observed in our killing assays.

In conclusion, the presented experimental platform harbors the ability to discover sequence-related and -unrelated on- and off-target sequences and is uniquely placed for systematic interrogation of TCR-like antibody interactomes in the future. The presented strategy can complement high-throughput platforms to provide an effective way for understanding physiologically relevant antigen interactomes of clinical candidates. We anticipate that MS-based assays could guide

drug alterations during the maturation cycles of the drug prior to entry into human trials, supporting a physiologically relevant safety profile. Other approaches have successfully identified requirements and binding rules, including off-target sequences, for TCR and TCR-like molecules, however, neither study has yet demonstrated the confirmed off-target antigens to be presented in healthy tissue and, therefore, physiologically relevant. In contrast, our methodology further demonstrates the suitability to evaluate interactomes TCR-like antibodies directly in human tissues of physiological or pathological relevance. This possibility fundamentally expands and circumvents the limitations of existing strategies and prioritizes relevant off-targets for ex vivo validation.

Despite achieving high sensitivity and the detection and identification of the target peptide with an estimated abundance of 10th of copies, considerations of sample loss during immunoprecipitation suggest that the sensitivity of the assay may be lower than estimated here[34]. Furthermore, high amounts of both test antibody (hundreds of micrograms) and human tissues (tens of milligrams) are needed for these experiments. Importantly, state-of-the-art proteomics

approaches are a discovery technology which suffer from incomplete spectral assignments, and therefore may miss peptide identifications due to low abundance and/or unknown sequence modifications. Therefore, it will remain essential to combine mass spectrometry with complementary assessments of off-target reactivity for de-risking strategies. With a fast-expanding market of increasingly sensitive mass spectrometers and a miniaturization of the workflow, such inter-actomics assays could however be improved and optimized in the coming years and established as high-throughput platforms for systematic tissue screening.

We conclude that in silico predictions and biostructural modeling are to date not sufficient for de-risking. To overcome these limitations and to support the development of novel drug candidates, we propose that a functional interactome characterization can, if complemented with ex vivo testing in human primary cells and tissues, enable successful assessment and optimization of TCR-like antibody therapeutics, and ultimately minimize risks for patients.

## Methods

### Primary tissues and cell lines
Human liver samples were obtained from Cytes Biotechnologies (Spain). Human lung and colon samples were obtained from HTCR (Germany). Ethical approval was granted by the Ethics Commission of the Faculty of Medicine in the LMU (number 025-12) the Bavarian State Medical Association (number 11142), and the Ethical Committee of Hospital Universitari Mútua de Terrassa (number P22 002). Human PBMC were purchased from Lonza (Switzerland), and US collection of samples and commercial distribution is approved and reviewed on an annual basis by the Institutional Review Board, US, under the title: "Prospective collection of non-mobilized leukocytes via leukapheresis for research".

Hybridoma cell lines for pan-HLA antibody production (clone SPVL3) were grown in serum-free medium (17.5 ml of hybridoma mix [20 g peptone (Sigma), 25 g D-glucose (Sigma), 100 ml L-glutamine solution (200 mM, 100x Gibco/Invitrogen), 100 ml Penicillin strepto-mycin (100x Gibco/Invitrogen), 100 ml non-essential amino acids (100x, Gibco/Invitrogen), 150ul 2-mercaptoethanol (1% in dH$_2$0, Sigma)] in 500 ml Hybridoma SFM (Gibco/Invitrogen). Hybridoma cells were seeded at 8 Mio cells and maintained in bioreactor flasks CL350 (Thermo Scientific) according to the manufacturer's instruction.

The T2 cell line (DSMZ# ACC598) is a hybrid of B-lymphoblastoid cell line 721.174 and 8-azaguanine and an ouabain-resistant variant of T-lymphoblastoid cell line human ovarian epithelial–serous cancer cell line. Cells are TAP deficient and express only small amounts of HLA-A2 and no B5 antigen on their surface. The cells were cultured in RPMI containing 10% FCS and split every three to four days (to 0.6 mio/mL) before reaching confluence.

Jurkat NFAT cell line is T lymphocyte cell line (developed by Signosis) with transfected NFAT luciferase construct. The cells were cultured in RPMI containing 10% FCS and 0.2 µg/ml Hygromycin B. Cells were split every three to four days (to 0.5 mio/ml) before reaching confluence.

### Statistics & reproducibility
No statistical method was used to predetermine sample size, and the number of replicates was chosen on the basis of material availability and feasibility. However, three replicate analyses were considered a minimum requirement. No data were excluded from the analyses. The experiments were not randomized. The Investigators were not blinded to allocation during experiments and outcome assessment. Rather, negative control samples were queued first for mass spectrometry acquisition in order to avoid chromatographic carryover.

### MAGE-A4 TCB generation: Generation of antigen and selection of specific antibodies by phage display
The DNA sequences of both the HLA-A*02:01 extracellular domain (position 25-299, Uniprot Q53Z42) and the beta-2-microglobulin (position 21-119, Uniprot P61769) were synthesized and inserted in frame into a prokaryotic recipient vector. A 2-fold G4S linker was inserted between the protein fragment and an N-terminal Avi-tag and His-Tag allowed specific biotinylation during co-expression with BirA biotin ligase.

Recombinant soluble HLA alpha chain (HLA-A*02:01) and beta2-microglobulin were isolated from *E. coli* as inclusion bodies. Additionally, the HLA chain was in vivo biotinylated in *E. coli* by co-expression of the enzyme biotin ligase BirA. The solubilized proteins were refolded together with the peptide using a renaturation process and subsequent purification using standard chromatographic procedures. Peptides used were derived from the human MAGE-A4 protein (GVYDGREHTV, residues 230-239), and as a tool for pre-clearing a peptide derived from the human WT1 protein (VLDFAPPGA, residues 37-45). Anti-HLA-A2/MAGA4$_{230-239}$ specific Fabs were selected by phage display from synthetic Fab libraries consisting of VL and VH pairings derived from different V-domain families. More precisely, clone 57D03 was selected from antibody library DutaLibTM-4H. The library is built by diversification of the same human antibody scaffold, Dutalys Dummy34.1. This scaffold comprises human VH3 consensus and human VK1 consensus framework region sequences, and CDR sequences that are similar to human VH3-23 and human VK1-O2 germline sequences.

Selection rounds (bio panning) were performed in solution according to the following pattern: (i) pre-clearing of ~ $10^{12}$ phagemid particles per library pool on neutravidin coated 96 well plates coated with 500 nM of an unrelated biotinylated HLA-A2/ WT1$_{37-45}$ complex, (ii) incubation of the non-HLA-A2/ WT1$_{37-45}$-binding phagemid particles with 100 nM biotinylated HLA-A2/MAGA4$_{230-239}$ complex for 0.5 h in a total volume of 800 µl, (iii) capture of biotinylated HLA-A2/ MAGA4$_{230-239}$ and specifically binding phage by adding 80 µl of streptavidin-coated magnetic particles for 20 min on a shaker, (iv) washing of respective magnetic particles 5-10x with 1 ml PBS/Tween20 and 5-10x with 1 ml PBS using a magnetic particle separator, (v) elution of phage particles by addition of 1 ml 100 mM TEA (triethylamine) for 5-10 min and neutralization by addition of an 1/2 volume of 1 M Tris/ HCl pH 7.4, (vi) re-infection of log-phase E. coli TG1 cells with the eluted phage particles, incubation on a shaker at 37 °C for 0.5 h, infection with helper phage VCSM13, incubation on a shaker at 30 °C over night and subsequent PEG/NaCl precipitation of phagemid particles to be used in the next selection round. Selections were carried out over 3 to 4 rounds using decreasing antigen concentrations of 100 nM, 50 nM, 50 nM and 10 nM, respectively.

### MAGE-A4 TCB generation: ELISA for characterization Fabs obtained by phage display
In case of the TM-4H, after selection round 3 and 4, part of the reinfected bacteria were used to prepare a dense TG1 *E. coli* culture was prepared and incubated 7-16 h at 30 °C. Following preparation of the polyclonal plasmid mini-prep, all of the Fab clones in the mini-prep, jointly representing the selection round output, are re-formatted in batch for soluble Fab fragment (sFab) expression, to enable screening ELISAs. The first step in this re-formatting process was to excise the Gene3 stump CDS contained in the selected Fab constructs in a restriction digest. The restriction digest was performed either using BamHI, in order to obtain soluble Fabs tagged with a T7 sequence. The resulting linear fragment was gel-purified, circularized in a standard ligation reaction and transformed into competent TG1 *E. coli* cells. Transformants were plated onto 2xYT agar or supplemented with 100 µg/ml Carbenicillin and 2% w/v glucose.

Individual clones were bacterially expressed as 1 ml cultures in 96-well format and supernatants were subjected to a screening by ELISA. Specific binders were defined as signals higher than 5 × background for HLA-A2/MAGA4$_{230-239}$ and signals lower than 3 x background for HLA-A2/ WT1$_{37-45}$. More precisely, Thermo Fisher neutravidin 96 well strip plates were coated with 10 nM of biotinylated HLA-A2/MAGA4$_{230-239}$ complex or 50 nM of biotinylated HLA-A2/ WT1$_{37-45}$ complex at 37 °C for 30 min followed by blocking of the plate with 2% MPBS (200 μl/well) for 1-2 h at room temperature. The plate was washed 3 times with PBS, then Fab containing bacterial supernatants were added and the plate was incubated at room temperature for 1 h. After another 3 washing steps with PBS, anti-FLAG-HRP secondary antibody (1:4000) or anti-T7-HRP secondary antibody (1:10000) was added and the plate was incubated for 1 h at room temperature. The plate was washed 3 times with PBS and developed by adding 100ul/well BM Blue POD (Roche). The enzymatic reaction was stopped by adding 50ul/well 1 M H2SO4. The OD was read at 450 nm (reference at 900 nm) for a final read-out of OD450-900. Clones specifically binding to HLA-A2/MAGA4$_{230-239}$ complex but not to biotinylated HLA-A2/ WT1$_{37-45}$ complex were sequenced and forwarded to IgG conversion for further characterization in cell activation assays.

## MAGE-A4 TCB generation: Cloning of variable antibody domains into expression vectors

All Fabs demonstrating specific binding to HLA-A2/MAGA4$_{230-239}$ complex by ELISA were converted into an IgG1/kappa antibody with PG LALA mutation on Fc. Therefore, the PCR-amplified DNA fragments of heavy and light chain v-domains were inserted in frame into either the human IgG1 constant heavy chain or the human constant kappa light chain containing respective recipient mammalian expression vector. The antibody expression was driven by an MPSV promoter and transcription was terminated by a synthetic polyA signal sequence located downstream of the CDS. In addition to the expression cassette, each vector contained an EBV oriP sequence for autonomous replication in EBV-EBNA expressing cell lines.

## MAGE-A4 TCB generation: Surface plasmon resonance for affinity characterization of IgG and TCB to HLA-A2/MAGA4$_{230-239}$ complex and CD3 epsilon

Specific binders identified by cell activation assays were characterized for their binding properties by surface plasmon resonance-screening (SPR) of human IgG PGLALA or T cell bispecific molecule using a Biacore T200 biosensor.

For characterization of the affinity towards the peptide-HLA complex, SPR experiments were performed on a Biacore T200 with HBS-EP+ as running buffer (0.01 M HEPES, 0.15 M NaCl, 0.003 M EDTA and 0.05% v/v Surfactant P20 pH 7.4). An anti-human Fc specific antibodies (GE Healthcare BR-1008-39) was immobilized by standard amine coupling chemistry on a CM5 sensor chip (GE Healthcare). The IgG constructs were captured for 60 s at a concentration of 2.5 nM with 10 μl/min flow rate. The temperature of the measurement was set to 25 °C. A three-fold dilution series of the refolded HLA-A2/MAGA4$_{230-239}$ complex (6.17, 18.5, 55.6, 167, 500, 1500 nM) was injected over the ligand at 30 ml/min for 240 s to record the association phase. The dissociation phase was monitored for 240 s at a flow rate of 30 μl/min and triggered by switching from the sample solution to HBS-EP + . The chip surface was regenerated after every cycle using two injections of 3 M MgCl2 for 30 s. Bulk refractive index differences were corrected by subtracting blank injections and by subtracting the response obtained from the reference flow cell without captured IgG. The affinity constants were derived from the kinetic rate constants by fitting to a 1:1 Langmuir binding using the Biacore evaluation software (GE Healthcare). The measure was conducted in triplicate with independent dilution series.

For characterization of affinity towards CD3 epsilon SPR experiments were performed on a Biacore T200 with HBS-P as running buffer (0.01 M HEPES, 0.150 M NaCl pH 7.4, 0.05% v/v Surfactant P20). An anti-PGLALA IgG capturing antibody was immobilized by standard amine coupling chemistry on a Series S C1 sensor chip (GE Healthcare). The TCB constructs were captured for 30 s at a concentration of 2.5 nM with 5 μl/min flow rate. The temperature of the measurement was set to 25 °C and 37 °C, respectively. A three-fold dilution series of human CD3ε (7.4, 22.2, 66.7, 200 and 600 nM) was injected over the ligand at 30 ml/min for 600 s to record the association phase. The dissociation phase was monitored for 900 s at a flow rate of 30 μl/min and triggered by switching from the sample solution to HBS-P. The chip surface was regenerated after every cycle using one injection of 10 mM NaOH for 60 s. Bulk refractive index differences were corrected by subtracting blank injections and by subtracting the response obtained from the reference flow cell without captured TCB. The affinity constants were derived from the kinetic rate constants by fitting to a 1:1 Langmuir binding using the Biacore evaluation software (GE Healthcare). The measure was conducted in triplicate with independent dilution series.

## Targeted MS assay for validation of GVYDGREHTV HLA-presentation in A375 cells

Cell pellets were lysed in a hypotonic buffer (Hepes 50 mM pH7) containing 1% non-ionic detergent (NP40) and protease inhibitors. Xenografts tissues have been homogenized in the same buffer using a Potter-Elvehjem tissue grinder. Lysates have been cleared by centrifugation. After immunoprecipitation of HLA-A2 molecules with biotinylated mouse anti-HLA-A2 (clone BB7.2; Biolegend) antibody, protein-peptides complexes were purified on streptavidin cartridges using the AssayMAP Bravo automated platform (Agilent). HLA-associated peptides were eluted and purified with reverse phase cartridges using the AssayMAP Bravo automated platform (Agilent). Eluted peptides were lyophilized using an Eppendorf Concentrator. Lyophilized peptides were resuspended in 2% acetonitrile and 5% formic acid solution, containing an isotopically labeled GVYDGREHTV [U-13C5,15N-Val] reference peptide. Peptide composition was analyzed by liquid chromatography (nano capillary system, Dionex Corporation, Sunnyvale, CA, USA) on a C18 reversed-phase nano-high-performance liquid chromatography column connected to a mass spectrometer (Q-Exactive HF-X, Thermo, CA, USA) via electrospray ionization (LC−ESI−MS/MS). Total AUC per peptide is the sum of areas under the curve for three [GVYDGREHTV] mass spectrometric transitions from the doubly charged precursor ion (m/z: 569.7796): Y [y8] ion (m/z = 976.4483); D [y7] (m/z = 813.3850); G [y6] (m/z = 698.3580). Complex quantification has been calculated as the average ratio of endogenous [GVYDGREHTV] peptide area under the curve (AUC) by comparison to spiked-in isotopically labeled [GVYDGREHTV] peptide AUC. The estimated copy number of endogenous [GVYDGREHTV] per cell is calculated by comparison to a spiked-in isotopically labeled [GVYDGREHTV] peptide, assuming 100% recovery for both standard and endogenous peptides at the end of the workflow.

## Interactome assay: HLA-ligand purification

All TCR-like TCB antibodies were purified from transiently transfected HEK293-EBNA or CHO cells via protein A affinity chromatography and size exclusion chromatography[35–37]. HLA-class II antibody was purified from hybridoma supernatants (clone SPVL3) using Protein A/G Sepharose beads (GE healthcare). Cross-linking of purified antibodies was performed as follows: 500 μg of antibody per 0.25 ml Protein A/G-bead mixture was incubated for 1 h at 4 °C and then washed with 5 ml borate buffer (0.05 M boric acid, 0.05 M KCl, 4 mM NaOH, pH 8.0), and cross-linked with 5 ml of 40 mM dimethyl pimelimidate in borate buffer. After 30 min, crosslinking was terminated by addition of 5 ml ice-cold 0.2 M Tris pH 8.0 and the beads were washed three times with 5 ml 50 ml Tris pH 8.0. 100 mg of xenograft material or healthy tissue

were lysed in a Precellys Evolution tissue homogenizer (Bertin Instruments) in 1 ml lysis buffer (0.5% (v/v) IGEPAL 630, 50 mM Tris pH 8.0, 150 mM NaCl and 1 tablet cOmplete Protease Inhibitor Cocktail EDTA-free (Roche) per 10 ml buffer) at 4 °C. Lysate clarification was achieved by centrifugation at 3000 x g for 10 mins followed by a 20,000 × g spin step for 60 min at 4 °C. Lysates were pre-cleared with 1.5 ml of Protein A/G-bead mixture, and then incubated with antibody beads for at least 5 h at 4 °C with gentle rotation. The antibody-resin was then washed sequentially with 5 ml of 50 mM Tris pH 8.0 containing 150 mM NaCl, 450 mM NaCl and lastly, no salt. Peptides were eluted from the captured HLA-complexes by addition of 1.5 ml of 10% acetic acid and dried by vacuum centrifugation. Peptides were further purified by solid phase extraction (Oasis columns, Waters) following the manufacturer's instructions. Further purification from larger protein material was achieved through filtering through a MWCO filter (Ultrafree-MC-PLHCC (5 K) Filter, Millipore). Purified peptides were then dried and resuspended in 20 µl of loading buffer (1%ACN, 0.1% TFA in water) for LC-MS analysis.

### Interactome Assay: Liquid chromatography - mass spectrometric (LC-MS) acquisition and analysis

All experiments were performed in triplicates. Peptides were analyzed by an Ultimate 3000 HPLC system coupled to a high field Q-Exactive (HFX) Orbitrap mass spectrometer (Thermo Scientific). Peptides were initially trapped in loading solvent, before RP separation with a 60 min linear acetonitrile in water gradient of 2-25% across a 75 µm × 50 cm PepMap RSLC C18 EasySpray column (Thermo Scientific) at a flow rate of 250 nl/min. Gradient solvents contained additional 1%(v/v) DMSO and 0.1%(v/v) formic acid. An EasySpray source was used to ionize peptides at 2000 V, and peptide ions were introduced to the MS at an ion transfer tube temperature of 305 °C. Ions were analyzed by data-dependent acquisition. Initially a full-MS1 scan (120,000 resolution, 60 ms accumulation time, AGC $3 \times 10^6$) was followed by 20 data-dependent MS2 scans (60,000 resolution, 120 ms accumulation time, AGC $5 \times 10^5$), with an isolation width of 1.6 m/z and normalized HCD energy of 25%. Dynamic exclusion was set for 30 s.

Raw data files were analyzed in Progenesis QI for Proteomics version 3.0 (Nonlinear Dynamics, Waters), and spectral identification was performed with PEAKS version XPro software (Bioinformatic Solutions) using a protein sequence fasta file containing 20,398 entries from 29.09.2021 with a 1% FDR (A375 xenograft datasets), or PEAKS version X using 20,397 reviewed human Uniprot entries downloaded on 04.03.2020 with a score cutoff of -10lgP = 20 resulting in an acceptable FDR below 5.5%. No enzyme specificity was set, peptide mass error tolerances were set at 5 ppm for precursors and 0.03 Da for MS2 fragments. Significant changes between samples and in peptide abundance were calculated using the inbuild ANOVA evaluation in Progenesis (p≤0.01 for all experiments) set in a between-subjects-design which uses variance stabilization by arcsinh normalization. Quantitative values for the individually measured molecule charge states are integrated for each peptide identification. NetMHCpan 4.1[38] was used to predict HLA peptide sequences to the regarding alleles present in the A375 cell line or tissues. The minimal predicted rank score was chosen to assign the likely HLA allele of origin (for A375 only), and binding was assigned for all peptides with a rank score ≤2 (all samples). Sequence logos were generated by Seq2logo2.0[39] and GibbsCluster2.0[40].

### Jurkat NFAT activation assay

TCB antibody dilutions were prepared in RPMI assay medium. A dilution row of 7 dilutions (1:10) was prepared by transferring and mixing 10 µL (starting with 110 µL/well in row A) to the subsequent wells containing 100 uL/well of RPMI media. T2 cells were washed with PBS and their viability was determined by Trypan Blue stain using EVE cell counter (＞90%). T2 cells were pulsed with peptides (cells were

incubated in 1 ml media at a density of 1 mio/ml in IMDM 10% FCS in the presence of 1×10-4 M – 1×10-9 M POTP or control RMFPNAPYL peptide for 2 h at 37 °C and washed with IMDM 10% FCS). Cells were resuspended in 2.5 ml IMDM 10% FCS to obtain a cell density of 400 000 cells per ml and 50 µl of this cell suspension were plated per well of a 96-well U- bottom well plate. This corresponds to 20 000 target cells per well. Jurkat NFAT effector cells were counted (their viability was >90% (determined by Trypan Blue stain using EVE cell counter (NanoTek)) and adjusted to 2.0 Mio/ml in RPMI 10% FCS. 50 µl of the cell suspension (100 000 cells) were transferred to the wells of the assay plates, containing TCB and target cells to final E:T ratio of 5:1.

100 µl of the antibody dilutions or medium were added per well, followed by 100 000 Jurkat NFAT effector cells and 20 000 target cells. The final E:T was 5:1 and the final volume per well was 200 ul. Assay plates were covered with lids, and incubated for 6 h for endpoint measurements, at 37 °C in a humidified CO2. Roughly 1 h before the readout, the appropriate number of frozen aliquots of prepared ONE-Glo solution was thawed at room temperature. The assay plates were taken out of the incubator and 100 µl were removed from each well. ONE-Glo solution was added in equivalent volumes (1:1 v/v of substrate solution and total assay medium per well; here either 100 µl ONE-Glo per well). The wells were mixed thoroughly by pipetting up and down and the mixture was transferred into white plates, which were then incubated for 10 min at room temperature in the dark, while shaking to allow lysis of target cells. Luminescence was measured using Perkin Elmer (for 96 well plates).

### T cell-dependent lysis assay

Co-cultures of HLA-A*02:01 human primary cells/organoids/organs-on-chips and allogenic peripheral blood mononuclear cells (PBMC) from healthy donors (purchased at Lonza and StemCell Technologies) were exposed to increasing concentrations of MAGE-A4 TCB and in vitro cytotoxicity evaluated with the most appropriate readout in all systems at 72 h. The assays were controlled by testing co-cultures upon incubation in medium only. The negative control DP47 (a non-tumor targeted, CD3 TCB) and the positive control ESK1 TCB (a non-selective HLA-A*02:01/WT1 molecule targeting multiple peptide-HLA-A*02:01 complexes, also referred to as ESK1) were assayed side by side with MAGE-A4 TCB.

For the lung, the AXLung-On-Chip constituted of alveolar type 1 and type 2 epithelial cells isolated from lung tissue and seeded on a porous flexible membrane to mimic the alveolar barrier was used as described[41,42]. After verification of the HLA-A2 haplotype, allogenic PBMCs prior stained with a proliferation dye (CellTrace #C34564, Life technologies) were added to the chip and apoptosis measured by live incubation with a Caspase 3/7 fluorescent probe (CellEvent #C34564, Thermofisher) later imaged by microscopy (EVOS M700) and analyzed with the Image J software.

For the liver, liver spheroids were generated with same-donor primary human hepatocytes and liver non-parenchymal cells[43] and pre-characterized for HLA-A2 haplotype (provided by Cytes Biotechnologies (Spain) or Lonza (Switzerland)). The spheroids were treated with antibodies in co-culture with allogenic PBMC. Hepatotoxicity was assessed by monitoring Aspartate Transaminase (AST) activity using commercially available test kits (Bayer EC 1.1.1.27) on an ADVIA 1650 autoanalyzer (Bayer HealthCare AG) according to the manufacturer's instructions.

For the intestine, duodenum- and colon-derived organoids were expanded from primary patient intestinal samples[44] and grown in differentiation conditions using commercially available media (#06010, #100-0212, #100-0191, Stemcell Technologies). Once reached maturation, organoids were passaged and cultured with allogenic PBMC. The day after, treatments were added together with a Caspase 3/7 fluorescent probe (CellEvent #C34564, Thermofisher). Co-cultures were imaged using the Operetta High-Content Imaging

System (PerkinElmer) and organoid apoptosis later quantified by measuring the total fluorescence signal of the Caspase-3/7 Green Detection Reagent per well (ImageJ software).

## Molecular modeling

All molecular modeling and structural analyses were performed with BIOVIA Discovery Studio version 2021. The RMFPNAPYL peptide in the X-ray structure with PDB ID 4WUU was mutated in silico to the sequence of RBM4B$_{59-67}$ (KLHGVNINV). To resolve steric clashes between residue Asn$_6$ of the peptide and residue His$_{74}$ of the HLA α-chain and residue Ile$_7$ of the peptide and residue Trp$_{147}$ of the HLA α-chain, we performed an energy minimization using the CHARMM force field and the GBSW implicit solvent model. During the energy minimization, we restrained the movement of all HLA α-chain and β$_2$ microglobulin atoms.

## Reporting summary

Further information on research design is available in the Nature Portfolio Reporting Summary linked to this article.

## Data availability

The mass spectrometry proteomics data have been deposited to the ProteomeXchange Consortium via the PRIDE partner repository with the following identifiers: MAGE-A4 TCR-like antibody immunoprecipitation in A375 and A375 MAGE-A4 KO xenograft tissue data are available via ProteomeXchange with identifier PXD048298. MAGE-A4 and ESK1 TCR-like antibody immunoprecipitation data in liver samples are available with identifier PXD048294. MAGE-A4 antibody immunoprecipitations in lung and colon tissue are available with identifier PXD048295. Source data are provided with this paper.

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

## Acknowledgements

We thank Prof. Christian Brander and Dr. Christoph T. Berger for critical review of the manuscript. The Hybridoma clone SPVL3 was a gift from Prof. Anthony Purcell. This project, awarded to NT, was funded by Roche Pharma.

## Author contributions

E.M.B. and N.T. conceptualized and led the project. A.N., A.A. and E.L. performed MS experiments. V.K. performed JNFAT assays. A.N., A.A., V.K., H.L., I.W., A.B. and N.T. performed analyses and data interpretation. L.J.H. and A.C. designed the MAGE-A4 TCB and performed the affinity testing experiments presented in this manuscript. E.M.B., P.U., C.K. and N.T. provided funding and resources. For the development of cellular models utilized in the manuscript, the following authors contributed as listed: Investigation: T.W., E.D., N.S., G.R.; Methodology: T.W., R.N., G.R.; Data Curation: D.O.F.; Conceptualization: E.B.N., J.S., N.H., L.C., C.K.; Supervision: N.H., J.S., L.C., C.K.; Formal Analysis: T.W., E.D., N.S., G.R., L.C.; Visualization: L.C.. E.M.B., N.T. and A.B. wrote the manuscript. E.M.B., L.C., C.K. and N.T. contributed to the manuscript review and rebuttal.

## Competing interests

All authors listed with Roche Pharma affiliation are directly employed by Roche Pharma. T.W., P.U., and C.K. are inventors on related patents. P.U. and C.K. declare stock ownership. N.T. has been a paid consultant to Roche Pharma. The remaining authors declare no competing interests.
