## [Peer Review File · Nature Communications]

The physiological interactome of TCR-like antibody therapeutics in human tissuesREVIEWER COMMENTS

Reviewer #1 (Remarks to the Author):

Review of Marrer-Berger et al “A de-risking experimental framework defines the physiological interactome of TCR-based therapeutics in human tissues”

General comments:

The manuscript describes a method to detect ligands for TCR-like antibodies. The antibodies were immobilized to Protein A/G beads. After crosslinking, the beads were incubated with lysates from cell lines, tumor xenografts, or primary healthy tissue. HLA-bound peptides were eluted with 10% acetic acid, purified through a 5kDA filter and analyzed by mass spectrometry.

The principle is known as immunoprecipitation and mass spectrometry (IP-MS). This is already a cornerstone in so-called immunopeptidomics where the aim is to identify all HLA-binding peptides. The novel aspect here is that the antibody used for IP recognizes a particular HLA-peptide complex rather than the HLA-molecule itself.

The authors used IP-MS to identify ligands for a novel TCR-like antibody targeting a peptide from MAGE-A4 bound to HLA-A02:01. The intended target was found to be the only peptide that was detected in two independent biological repeat experiments. An antibody targeting an irrelevant HLA-peptide complex was used as control.

IP-MS was next used to identify cross-reactive peptides bound by an antibody (ESK1) that has previously been shown to be cross-reactive. The ESK1 antibody targets a peptide from WT-1 bound to HLA HLA-A02:01. An earlier study had identified 100 peptides using a mini-gene approach (PresentER). Here, the authors identified 16 peptides from primary liver tissue. Half of these were shown to induce T-cell activation, but there was no correlation between T-cell activation and signal intensity in the ligand detection assay (IP-MS). There was also no overlap between the peptides detected here and those identified in the mentioned earlier study.

Specific comments:

De novo identification of TCR-ligands is a major challenge in immunology and immunotherapy. Yet, the method described was not applied for TCRs but rather to TCR-like antibodies. It seems unlikely that the same approach can be used to identify TCR-ligands due to the low affinity of TCR-HLAp interactions. The method is therefore likely to be limited to identifying ligands for TCR-like antibodies, making the title misleading.

The results in Fig. 2A and 5A show that the number of peptides identified by the method can be very high. Yet, there is no information about reproducibility in the text or the figure legends. For Fig 5, there is also discrepancy between the main text and the figure legend explaining the use of an isotype-control (anti-HLA-DQ) or ESK1, respectively. The axes read "MAGE/CTRL". Moreover, the authors explain that the intended target for the MAGE4-HLA-A2 antibody was the only peptide that was detected in two biological replicates. This indicates that there is considerable variation between results obtained in replicates.

The title suggests that the intention is to de-risk a candidate therapeutic. The question is therefore if the method can replace testing of T-cell reactivity. This would be an important achievement since testing T-cell reactivity against primary tissue is often not possible. However, to prove this, one would need more data. To start with I would suggest testing a panel of TCR-like antibodies with different reactivity against a panel of cell lines in cell-based activation assays. One can then determine to what extent IP-MS is predictive of reactivity in cell-based assays.

Reviewer #2 (Remarks to the Author):

The work by Marrer-Berger and colleagues addresses a major, unmet need for the development of pHLA-targeted therapeutics: assessment of cross-reactivities with self-antigens to identify and mitigate safety risks. The authors outline a reverse immunopeptidomics platform to effectively identify cross-reactivities for endogenous peptides, and apply it to evaluate and compare the specificity profiles of two therapeutic antibodies in bi-specific T cell engager format, a previously developed ESK1 which targets

WT-1 epitope and a novel TCR-like Ab developed in this work (57D03) which recognizes the known MAGE-A4 epitopic peptide presented on HLA-A*02:01. Conclusions from the analysis can already aid the development of pHLA-targeted immunotherapies and provide a rationale that could greatly benefit similar efforts that are currently underway by several groups in the immuno-oncology space.

The authors apply a real tour de force of different peptidomic and immunological assays, using extensive analysis done both xenograft models, primary human tissues, and ultimately activation assays using Jurkat cells and 2D liver spheroids, to support their conclusions. The work is technically sound, with proper experimental controls and rigorous evaluation of statistical significance of the results. Finally, the manuscript is written in a clear and succinct manner for a general audience.

I believe this work can be appropriate for publication in Nature communications, once the following important points have been adequately addressed by the authors:

Major comments:

1. What is the % recovery of eluted peptides from the HLAs purified using the process developed by the authors? In order for their method to be widely applicable as a tool in drug development, this must be established first. While exhaustiveness seems nearly impossible to accomplish, some estimate of the % coverage of endogenous peptidomes should be provided.
2. In addition to the work by Gejman and colleagues that is extensively discussed and compared with the present work, previous platforms have been described to evaluate cross-reactivities between TCR-like molecules and pHLA antigens (see, for example Birnbaum et al., Cell 2014; Gee et al., Cell 2018; Kula et al., Cell, 2019). These methods claim some level of exhaustiveness in identifying pHLA antigens for specific TCRs, so it would be important to outline how the current approach compares to these methods for a specific exemplar antigen.

3. How sensitive is the interaction between the 57D03 with pHLAs presenting MAGE-A4 on the HLA allotype? This should be explicitly addressed by evaluating cross-reactivities with other allotypes that can display the same peptide, either in vitro or even better in situ using the author's novel technology.

4. The structural modeling and subsequent SASA/RMS analyses presented in Figure 4 and Table 2, respectively, as a rationale for the molecular basis of the identified cross-reactivities assumes that all peptides are presented in a very similar backbone conformation, and that binding of ESK1 doesn't induce significant structural changes. For example, it is conceivable that peptides may adopt different backbone conformations presenting key hotspot residues in geometries which mimic the Ab-bound complex, or that the ESK1 CDRs undergo considerable structural adaptations to accommodate different antigens, in an induced-fit type mechanism. Both assumptions must be verified experimentally by the authors by determining the crystal structures of the pHLA/ESK1 complexes to adequately support their conclusions. In the absence of experimentally determined structures these conclusions are highly speculative.

Minor comments:

The primary SPR data (sensograms) summarized in Supp. Table 1 should be also provided, including the fitted curves which were used to derive on and off rates.

Since the novelty of the work lies primarily in the immunopeptidomic analysis, the primary data should be made available in an existing data base or author-supplied web-page.

Reviewer #3 (Remarks to the Author):

In this manuscript, Marrer-Berger and colleagues experimentally tested a workflow to assess the specificity of TCR-like molecules targeting tumor-derived peptides. In particular, they compared a homemade antibody targeting MAGE-A4 and a previously described antibody (ESK1) against the WT-1 antigen exhibiting off-target binding. Using Mass spectrometry analyses, they identified the peptides expressed in healthy tissues recognized

by these antibodies and demonstrated their potency to elicit T cell activation.

The results and methods used are interesting but some concerns need to be clarified to fully support the conclusions raised by the authors.

Figure 1D-F. The authors found that IP with Mage-A4 ab detects significant enrichment of the expected peptide compared to control IPs. Could the complete list of detected peptides be made available to readers? Are other peptides significantly enriched in at least one out of the two experiments?

Figure 2. The design of the experiment is quite confusing. The authors claim that “Since both target antigens MAGE-A4 WT-1 are not expressed in A375 cells (data not shown), the ESK1 enrichment could serve as isotype control for the MAGE-A4 IgG, and vice versa”. As used for previous experiments, A375 express MAGE-A4 IgG targets. To my knowledge, A375 cell line also express WT-1. in any case, it is difficult to understand why the expression of these antigens in the A375 line is a determining argument for the selection of control IPs made with liver tissue. The control IP should have been performed with an isotype control antibody.

Figure 3D. The authors claim that MAGE-A4 and WT-1 are not expressed by the liver spheroids used in the experiments. This aspect should be tested if it has not been evaluated in previous studies (cite as appropriate). The authors should display raw values rather than normalized values, to assess the background (generated by control TCB) of cell killing resulting from these experiments.

Figure 5A. What are the few hits that are enriched with MAGE-A4 IgG in the volcano plot Lung (tissue)? Although they are not HLA-A*20:01 bind peptides, they might constitute unrelated off-targets. As point out above, the authors should display raw values rather than normalized values, to assess the background (generated by control TCB) of cell killing resulting from these experiments (% of cells remaining compared to the starting number incubated for the assay).

Minor points:

1. The authors should be more precise about the molecules (antibody or TCB) they used in the different experiments. For example, Line 111, presumably the authors used "MAGE-A4 TCB" for this experiment.
2. line 119, correct Figure 1C for Figure 1D.
3. Figure 2 and 5, the authors should specify the number replicates performed to obtain the volcano plots and the p-value threshold selected for the determination of enriched peptides. The list of all detected peptides with their corresponding p-values and ratios should be available for the readers.
4. What is the threshold used to predict peptide binding with HLA-A*02:01, and what are the Scores obtained with NetMHCpan-4.1 for each detected peptide?
5. line 174, correct Figure 3B for Figure 3C
6. line 181, correct Figure 3C for Figure 3B
7. The legend of Figure 5A is incorrect

RESPONSES TO REVIEWER COMMENTS

(Authors responses in green, with citations from manuscript highlighted in italic)

Reviewer #1 (Remarks to the Author):

Review of Marrer-Berger et al "A de-risking experimental framework defines the physiological interactome of TCR-based therapeutics in human tissues"

General comments:

The manuscript describes a method to detect ligands for TCR-like antibodies. The antibodies were immobilized to Protein A/G beads. After crosslinking, the beads were incubated with lysates from cell lines, tumor xenografts, or primary healthy tissue. HLA-bound peptides were eluted with 10% acetic acid, purified through a 5kDA filter and analyzed by mass spectrometry.

The principle is known as immunoprecipitation and mass spectrometry (IP-MS). This is already a cornerstone in so-called immunopeptidomics where the aim is to identify all HLA-binding peptides. The novel aspect here is that the antibody used for IP recognizes a particular HLA-peptide complex rather than the HLA-molecule itself.

The authors used IP-MS to identify ligands for a novel TCR-like antibody targeting a peptide from MAGE-A4 bound to HLA-A02:01. The intended target was found to be the only peptide that was detected in two independent biological repeat experiments. An antibody targeting an irrelevant HLA-peptide complex was used as control.

IP-MS was next used to identify cross-reactive peptides bound by an antibody (ESK1) that has previously been shown to be cross-reactive. The ESK1 antibody targets a peptide from WT-1 bound to HLA HLA-A02:01. An earlier study had identified 100 peptides using a mini-gene approach (PresentER). Here, the authors identified 16 peptides from primary liver tissue. Half of these were shown to induce T-cell activation, but there was no correlation between T-cell activation and signal intensity in the ligand detection assay (IP-MS). There was also no overlap between the peptides detected here and those identified in the mentioned earlier study.

Specific comments:

De novo identification of TCR-ligands is a major challenge in immunology and immunotherapy. Yet, the method described was not applied for TCRs but rather to TCR-like antibodies. It seems unlikely that the same approach can be used to identify TCR-ligands due to the low affinity of TCR-HLAp interactions. The method is therefore likely to be limited to identifying ligands for TCR-like antibodies, making the title misleading.

We thank the reviewer for this comment and agree that we cannot yet state whether this technology can be extended to recombinant TCR molecules. We have therefore adjusted the title in order to reflect that our current approach was applied to "*TCR-like antibody therapeutics*" rather than "TCR-based therapeutics".

We completely agree with the reviewer that the applicability of our methodology to native TCRs has yet to be tested. We have revisited the affinity measurements for the MAGE-A4 molecule at 25 degrees (please see amended Supplementary Figure 1 which is now including original measurement datasets), which is 16 nM. Therefore, it is anticipated that our experimental framework could be applicable to high affinity native TCRs, and affinity matured molecules.

The results in Fig. 2A and 5A show that the number of peptides identified by the method can be very high.

Yet, there is no information about reproducibility in the text or the figure legends. For Fig 5, there is also discrepancy between the main text and the figure legend explaining the use of an isotype-control (anti-HLA-DQ) or ESK1, respectively. The axes read "MAGE/CTRL". Moreover, the authors explain that the intended target for the MAGE4-HLA-A2 antibody was the only peptide that was detected in two biological replicates. This indicates that there is considerable variation between results obtained in replicates.

Since immunoprecipitation (IP) experiments enrich for target molecules, but are prone to co-precipitation, we do detect significant background signal in these experiments. This background arises from co-precipitation due to incomplete solubilisation of membrane, and secondary protein/protein interactions¹. Therefore, our presented strategy to subtract background signal is crucial for our highly specific results. We found that triplicate analyses are sufficient to address variability in the MS-based assays, and that we can shortlist functionally relevant peptides with high specificity when comparing the target IP to results obtained from a control IP using an isotype control. This procedure ensures removal of signal from non-specifically co-precipitating peptides and validates reproducible peptide enrichment. We validate this strategy using the ESK1 antibody, for which we observe that half of the peptides we shortlisted from triplicate analysis are confirmed functionally active. Such high specificity has not been achieved with any other methodology that aims to discover TCR target peptides, nor has any other methodology concluded in such a high number of physiologically relevant off-target antigens.

We have highlighted this in an additional sentence in the discussion, page 14: "*Such exceptionally high specificity has not been achieved by any other methodology.*"

Figure 5 shows data generated with SPVL3 as the control antibody, and there was an error in the figure legend which is now corrected. We sincerely apologise about this oversight. For clarity, we have stated which antibody was used as a control directly in the figure as well. We have carefully reviewed and extended the text where needed to improve clarity, and the according datasets have now been appended as supplementary data files.

We realized that the section mentioned above was not formulated clearly, and we have amended it as follows in the relevant result section on page 6: "*GVDYDGREHTV was the only sequence that was identified as significantly enriched by the MAGE-A4 antibody in A375 xenografts, indicating a highly selective binding capacity of the antibody (Supplementary table 1)*" As indicated, this data is now also available as Supplementary data table.

The title suggests that the intention is to de-risk a candidate therapeutic. The question is therefore if the method can replace testing of T-cell reactivity. This would be an important achievement since testing T-cell reactivity against primary tissue is often not possible. However, to prove this, one would need more data. To start with I would suggest testing a panel of TCR-like antibodies with different reactivity against a panel of cell lines in cell-based activation assays. One can then determine to what extent IP-MS is predictive of reactivity in cell-based assays.

We believe that our assay will be instrumental in contributing to the evaluation of cross-reactivity of TCR-like therapeutics, and therefore significantly enhance the ability to de-risk TCR-like therapeutics. But we see our methodology not as a standalone platform to address safety profiles, but as complimentary to existing pipelines. We anticipate that integrating MS screening approaches during the maturation of TCR-like therapeutics for applicability in the clinic will enable more physiologically relevant guidance on the maturation process, with benefits to the safety profile of these molecules for clinical application. We have highlighted this now accurately in the title ("de-risking" removed, and title amended to: "The physiological interactome of TCR-like antibody therapeutics") and reformulated our expectations in the manuscript.

Please also see our response to reviewer 2, question 1 in this context.

We also agree that, in principle, it would be highly beneficial to test a wider panel of TCR-like antibodies using our assay. However, the only other available bispecific molecule targeting a specific HLA-peptide

complex is Immunocore's GP100 molecule that we do not have access to in the amounts needed for these experiments). The so far limited availability of reagents highlights the importance and timeliness of our work, and we hope that this is acknowledged.

Since we agree with the author that we have not (yet) shown that this technology will support the de-risking of TCR-like antibodies, we have altered the title and manuscript text accordingly.

"Page 15: " The presented strategy can complement high-throughput platforms to provide an effective way for understanding physiologically relevant antigen interactomes of clinical candidates. We anticipate that MS-based assays could guide drug alterations during the maturation cycles of the drug prior to entry into human trials, supporting a physiologically relevant safety profile."

Reviewer #2 (Remarks to the Author):

The work by Marrer-Berger and colleagues addresses a major, unmet need for the development of pHLA-targeted therapeutics: assessment of cross-reactivities with self-antigens to identify and mitigate safety risks. The authors outline a reverse immunopeptidomics platform to effectively identify cross-reactivities for endogenous peptides, and apply it to evaluate and compare the specificity profiles of two therapeutic antibodies in bi-specific T cell engager format, a previously developed ESK1 which targets WT-1 epitope and a novel TCR-like Ab developed in this work (57D03) which recognizes the known MAGE-A4 epitopic peptide presented on HLA-A*02:01. Conclusions from the analysis can already aid the development of pHLA-targeted immunotherapies and provide a rationale that could greatly benefit similar efforts that are currently underway by several groups in the immuno-oncology space.

The authors apply a real tour de force of different peptidomic and immunological assays, using extensive analysis done both xenograft models, primary human tissues, and ultimately activation assays using Jurkat cells and 2D liver spheroids, to support their conclusions. The work is technically sound, with proper experimental controls and rigorous evaluation of statistical significance of the results. Finally, the manuscript is written in a clear and succinct manner for a general audience.

I believe this work can be appropriate for publication in Nature communications, once the following important points have been adequately addressed by the authors:

Major comments:

1. What is the % recovery of eluted peptides from the HLAs purified using the process developed by the authors? In order for their method to be widely applicable as a tool in drug development, this must be established first. While exhaustiveness seems nearly impossible to accomplish, some estimate of the % coverage of endogenous peptidomes should be provided.

We completely agree with the reviewer - the percentage of recovery is an important question and probably one of the most important questions in the field. % coverage has so far only been estimated by comparing the number of molecules known to be likely expressed on the surface of a tumour cell (100,000-500,000)², and the number of different HLA sequences recovered. Furthermore, an attempt has been made to estimate recoveries using stable isotope labelled HLA-peptide complexes³, concluding recovery is as low as 5% of total HLA material. Estimating even a minimum number of 1*E5 HLA molecules per cell, and an estimated minimal cell number of 5*10E8 from the xenograft material and/or tissue used, a peptide with a single copy number therefore would be recovered in this identical copy number, i.e. 5*10E8, equating to 160 attomole of peptide. However, with a 95% loss, we arrive at low attomoles for the least abundant peptides. Therefore, we conclude that we can measure peptide with an estimated copy number of 100 molecules/cell in our experiments.

In addition, proteomics technology can only confidently assign peptide sequences to a certain percentage of the measured spectra. This is due to (i) under-sampling, and (ii) unknown potential post-translational

modifications of the peptide molecules. Under-sampling means that peptide molecules that elute from the chromatography and are introduced to the mass spectrometer may be missed if too many other molecules are eluted at the same time, as every peptide is sequentially isolated, fragmented and measured within the mass spectrometer. Modifications on the other hand may make it difficult to assign the correct sequence to an acquired spectrum, making it more difficult to detect post-translationally modified peptides with this technology.

Therefore, we consider our approach and mass spectrometry in general a *discovery* technology, with value on the detected, and a factor of unknown in the undetected.

This is also why we see this approach as complimentary to other approaches for off-target identification. As we stated in the reply 3 to reviewer 1 (see above), we see our methodology not as a standalone platform to address safety profiles. We anticipate that integrating MS screening approaches during the maturation of TCR-like therapeutics for applicability in the clinic will enable more physiologically relevant guidance on the maturation process, with benefits to the safety profile of these molecules for clinical application. We have amended the manuscript and title accordingly, and have included these considerations in the discussion:

Page 15: "Despite achieving high sensitivity and the detection and identification of the target peptide with an estimated abundance of 10th of copies per cell, considerations of sample loss during immunoprecipitation suggest that the sensitivity of the assay may be lower than estimated here³. Furthermore, high amounts of both test antibody (hundreds of micrograms) and human tissues (tens of milligrams) are needed for these experiments. Importantly, state-of-the-art proteomics approaches still suffer from incomplete spectral assignments, and therefore may miss peptide identifications due to low abundance and/or sequence modifications. Therefore, it will remain essential to combine mass spectrometry with complementary assessments of off-target reactivity for de-risking strategies."

2. In addition to the work by Gejman and colleagues that is extensively discussed and compared with the present work, previous platforms have been described to evaluate cross-reactivities between TCR-like molecules and pHLA antigens (see, for example Birnbaum et al., Cell 2014; Gee et al., Cell 2018; physiological et al., Cell, 2019). These methods claim some level of exhaustiveness in identifying pHLA antigens for specific TCRs, so it would be important to outline how the current approach compares to these methods for a specific exemplar antigen.

These are all important references for known approaches to TCR specificity evaluation. In brief, Birnbaum *et al.*⁴, already cited in our manuscript, used a highly diverse library of >10E8 peptides, and used this to obtain a footprint of amino acid preferences within a 10mer antigen presented by MHC interacting with a specific TCR. Authors then predict from the canonical human proteome which sequences may be presented and match the TCR interaction profile.

This footprint will be very accurate, as it tests a high number of possible interactions with the TCR and provides an unbiased view on its interaction surface. However, this approach, including all other approaches to date, do not provide an understanding of which peptides are physiologically presented in cells and tissues - crucial information to understand causality of off-target reactions. Our approach on the contrary, offers direct identification of physiologically relevant off-target peptides.

We had previously missed to include Kula *et al.*, Cell 2019⁵, which we have now included in the list of references. The authors here use lentiviral libraries that insert a short ORF into EDCs expressing HLA-A2. Here, intracellular protein will increase the likelihood of the sequences presented by HLA to be physiologically meaningful. However, in this case, the size of the libraries tested are limited to much smaller size (reported here 2,882 peptides), and only a certain range of possible peptide sequences can be tested in detail.

Finally, Gee *et al.*, also cited in the manuscript, has the most comprehensive library yet (4E8) and again, despite the fact that this approach very accurately profiles the interaction surface of the TCR, it cannot provide causal target-reactivity unless otherwise validated.

Our approach automatically enriches and identifies naturally presented, physiologically relevant peptide sequences interacting with a TCR-like antibody in human tissues, which we believe is the unique capacity of our assay, and the most relevant advancement over existing methodologies in the field, which we have outlined in the discussion of the manuscript.

3. How sensitive is the interaction between the 57D03 with pHLAs presenting MAGE-A4 on the HLA allotype? This should be explicitly addressed by evaluating cross-reactivities with other allotypes that can display the same peptide, either in vitro or even better in situ using the author's novel technology.

We have now included detailed Biocore affinity measurement datasets in Supplementary Figure 1. It is an exciting suggestion to test TCR-like therapeutics for alloreactivity. We have not addressed this question in our current work and think this is out of scope of our current study as many of our assays are not established beyond HLA-A*02:01. We are very interested to pursue this in our future work.

4. The structural modeling and subsequent SASA/RMS analyses presented in Figure 4 and Table 2, respectively, as a rationale for the molecular basis of the identified cross-reactivities assumes that all peptides are presented in a very similar backbone conformation, and that binding of ESK1 doesn't induce significant structural changes. For example, it is conceivable that peptides may adopt different backbone conformations presenting key hotspot residues in geometries which mimic the Ab-bound complex, or that the ESK1 CDRs undergo considerable structural adaptations to accommodate different antigens, in an induced-fit type mechanism. Both assumptions must be verified experimentally by the authors by determining the crystal structures of the pHLA/ESK1 complexes to adequately support their conclusions. In the absence of experimentally determined structures these conclusions are highly speculative.

The reviewer is correct in pointing out that it has been shown that some peptides presented on HLA-A*02 are able to adopt diverse backbone geometries and that "register shifts" (occupation of the hydrophobic cavities by residues other than the canonical anchors) do occur (e.g., for the 10mer peptide MMWDRGLGMM in Riley *et al.*⁶).

When aligning the structures of 130 HLA-A*02/9mer peptide X-ray complex structures published between 1997 and 2021 (non-redundant with regard to the peptide sequences), the average peptide backbone C-alpha RMSD is only 0.86 Angström. The structural alignment of the 130 peptides documents a very uniform backbone conformation in the N-terminal part of the known crystallized 9mer peptides, i.e., the part of the peptide that likely would be in contact with ESK-1.

As "our" peptides all have a charged residue (R/K) on position 1 and a classic hydrophobic anchor residue (L/V) on position 2, we dare to suggest that a register shift is unlikely. The same is true for the C-terminal part, where there are polar or charged residues on position 8 (S/T/N/H/R), and a classical hydrophobic anchor residue on position 9 (L/V). Hence, the situation is markedly different from the published example MMWDRGLGMM, where the double "MM" motif at both termini (plus additional backbone degrees of freedom related to the peptide being a 10mer) are likely contributors to register shifting and the associated larger structural rearrangements.

Likewise, it is true that antibody CDRs are known to adopt multiple conformations. Apo and antigen-complexed Fab X-ray crystal structures of the same antibody often show structural differences in the CDR loops. The presence of multiple antibody paratope states has recently been modelled using MD simulation and Markov state models⁷. As the CDRs of ESK-1 have a sizable number of conformational degrees of freedom (CDR-H3 and CDR-L3 are both 11 amino acids long), it is conceivable that multiple paratope conformation exist, i.e., that ESK-1, to a certain degree, is able to adapt its conformation to the pMHC.

We still argue that these rearrangements, for "our" set of peptides, are going to be negligible. First, the ESK-1 epitope is, to a large part, consisting of HLA-A*02 residues. Within the 4 Angström range of the ESK-1 Fab in the X-ray complex structure with PDD ID 4wuu, there are 20 amino acids belonging to the HLA alpha chain, and only 2 residues belonging to the peptide (residues R1 and P4). Since a wide part of the ESK-1 epitope is invariant (non-peptide dependent), the potential for structural rearrangements is limited.

Second, same as the RMF peptide, all "our" peptides have a positively charged amino acid on position 1 (R/K). The positively charged amino acid on peptide position 1 is clearly the key interaction residue for ESK-1, being sandwiched between the CDR-H3 backbone, the sidechain of CDR-H3 residue Y100, and the sidechains of CDR-L3 residues W91 and D93. Essentially, ESK-1 does not need to rearrange as all its key interaction sites, when adopting the same binding mode as to HLA-A*02/RMF, are already in place.

Therefore, while we acknowledge the point that conformational flexibility is not exhaustively modelled in our structural assessment, we argue that our reasoning is not speculative but provides a sound explanation for the recognition of these pMHCs by ESK-1. Even if larger structural rearrangements were to occur, it would not de-validate the functionality of the approach.

Minor comments:

The primary SPR data (sensograms) summarized in Supp. Table 1 should be also provided, including the fitted curves which were used to derive on and off rates.

This is now provided, please see extended Supplementary Figure 1.

Since the novelty of the work lies primarily in the immunopeptidomic analysis, the primary data should be made available in an existing data base or author-supplied web-page.

We have now provided all quantitative datasets in the supplementary data. MS raw data will be made available upon request as stated in data availability.

Reviewer #3 (Remarks to the Author):

In this manuscript, Marrer-Berger and colleagues experimentally tested a workflow to assess the specificity of TCR-like molecules targeting tumor-derived peptides. In particular, they compared a homemade antibody targeting MAGE-A4 and a previously described antibody (ESK1) against the WT-1 antigen exhibiting off-target binding. Using Mass spectrometry analyses, they identified the peptides expressed in healthy tissues recognized by these antibodies and demonstrated their potency to elicit T cell activation.

The results and methods used are interesting but some concerns need to be clarified to fully support the conclusions raised by the authors.

Figure 1D-F. The authors found that IP with Mage-A4 ab detects significant enrichment of the expected peptide compared to control IPs. Could the complete list of detected peptides be made available to readers? Are other peptides significantly enriched in at least one out of the two experiments?

We have now provided the complete datasets for all experiments in the supplementary data.

Figure 2. The design of the experiment is quite confusing. The authors claim that "Since both target antigens MAGE-A4 WT-1 are not expressed in A375 cells (data not shown), the ESK1 enrichment could serve as isotype control for the MAGE-A4 IgG, and vice versa". As used for previous experiments, A375 express

MAGE-A4 IgG targets. To my knowledge, A375 cell line also express WT-1. in any case, it is difficult to understand why the expression of these antigens in the A375 line is a determining argument for the selection of control IPs made with liver tissue. The control IP should have been performed with an isotype control antibody.

We sincerely apologise for this confusion. The reviewer is of course completely correct that MAGE-A4 and WT1 are expressed in A375 cells, which is our model cell line in which we have confirmed detection of the MAGE-A4 target peptide. The result section we stated this falsely in is when we applied both antibodies to IP potential off-targets in healthy liver tissue, which does not express either antigen. This error has now been corrected.

Figure 3D. The authors claim that MAGE-A4 and WT-1 are not expressed by the liver spheroids used in the experiments. This aspect should be tested if it has not been evaluated in previous studies (cite as appropriate). The authors should display raw values rather than normalized values, to assess the background (generated by control TCB) of cell killing resulting from these experiments.

It is established that MAGE-A4 and WT-1 are not expressed in liver tissue, and we have now cited the Human Protein Atlas accordingly in the manuscript.

Figure 5A. What are the few hits that are enriched with MAGE-A4 IgG in the volcano plot Lung (tissue)? Although they are not HLA-A*20:01 bind peptides, they might constitute unrelated off-targets. As point out above, the authors should display raw values rather than normalized values, to assess the background (generated by control TCB) of cell killing resulting from these experiments (% of cells remaining compared to the starting number incubated for the assay).

All quantitative data tables have now been made available as supplementary data.

Minor points:

1. The authors should be more precise about the molecules (antibody or TCB) they used in the different experiments. For example, Line 111, presumably the authors used "MAGE-A4 TCB" for this experiment. This is correct, and we have amended this accordingly and reviewed the manuscript carefully again for such errors. We also introduced how we refer to either TCB or antibody separately.

Page 5 line 108: "... HLA-A2 MAGE-A4 targeting TCR-like TCB (henceforth referred to as MAGE-A4 TCB..." and Page 6, line 126: "... IgG-format of the MAGE-A4 TCR-like antibody (henceforth referred to as MAGE-A4 antibody) ..."

2. line 119, correct Figure 1C for Figure 1D. Done.

3. Figure 2 and 5, the authors should specify the number replicates performed to obtain the volcano plots and the p-value threshold selected for the determination of enriched peptides. The list of all detected peptides with their corresponding p-values and ratios should be available for the readers.

Supplementary Data has now been added accordingly for all experiments.

4. What is the threshold used to predict peptide binding with HLA-A*02:01, and what are the Score obtained with NetMHCpan-4.1 for each detected peptide?

The recommended threshold of rank score ≤ 2 was used as a cut-off score for binding. This has now been added to the materials and methods section as follows:

Page 23, line 529: "*NetMHCpan 4.1 (services.healthtech.dtu.dk) was used to predict HLA peptide sequences to the regarding alleles present in the A375 cell line or tissues. The minimal predicted rank score was chosen to assign the likely HLA allele of origin, and binding was assigned for all peptides with a rank score ≤ 2 .*"

5. line 174, correct Figure 3B for Figure 3C Panels have now been switched in figure accordingly

6. line 181, correct Figure 3C for Figure 3B Panels have now been switched in figure accordingly

7. The legend of Figure 5A is incorrect. Apologies, this is now corrected.

REFERENCES

1. Partridge, T., Nicastrì, A., Kliszczak, A.E., Yindom, L.M., Kessler, B.M., Ternette, N., and Borrow, P. (2018). Discrimination Between Human Leukocyte Antigen Class I-Bound and Co-Purified HIV-Derived Peptides in Immunopeptidomics Workflows. *Front Immunol* 9, 912. 10.3389/fimmu.2018.00912.
2. Yewdell, J.W., Reits, E., and Neefjes, J. (2003). Making sense of mass destruction: quantitating MHC class I antigen presentation. *Nat Rev Immunol* 3, 952-961. 10.1038/nri1250.
3. Hassan, C., Kester, M.G., Oudgenoeg, G., de Ru, A.H., Janssen, G.M., Drijfhout, J.W., Spaapen, R.M., Jimenez, C.R., Heemskerk, M.H., Falkenburg, J.H., and van Veelen, P.A. (2014). Accurate quantitation of MHC-bound peptides by application of isotopically labeled peptide MHC complexes. *J Proteomics* 109, 240-244. 10.1016/j.jprot.2014.07.009.
4. Birnbaum, M.E., Mendoza, J.L., Sethi, D.K., Dong, S., Glanville, J., Dobbins, J., Ozkan, E., Davis, M.M., Wucherpfennig, K.W., and Garcia, K.C. (2014). Deconstructing the peptide-MHC specificity of T cell recognition. *Cell* 157, 1073-1087. 10.1016/j.cell.2014.03.047.
5. Kula, T., Dezfulian, M.H., Wang, C.I., Abdelfattah, N.S., Hartman, Z.C., Wucherpfennig, K.W., Lyerly, H.K., and Elledge, S.J. (2019). T-Scan: A Genome-wide Method for the Systematic Discovery of T Cell Epitopes. *Cell* 178, 1016-1028 e1013. 10.1016/j.cell.2019.07.009.
6. Riley, T.P., Hellman, L.M., Gee, M.H., Mendoza, J.L., Alonso, J.A., Foley, K.C., Nishimura, M.I., Vander Kooi, C.W., Garcia, K.C., and Baker, B.M. (2018). T cell receptor cross-reactivity expanded by dramatic peptide-MHC adaptability. *Nat Chem Biol* 14, 934-942. 10.1038/s41589-018-0130-4.
7. Fernandez-Quintero, M.L., Pomarici, N.D., Math, B.A., Kroell, K.B., Waibl, F., Bujotzek, A., Georges, G., and Liedl, K.R. (2020). Antibodies exhibit multiple paratope states influencing V(H)-V(L) domain orientations. *Commun Biol* 3, 589. 10.1038/s42003-020-01319-z.

REVIEWER COMMENTS

Reviewer #1 (Remarks to the Author):

A main concern with the original manuscript was that the authors describe the approach as a generic method to assess specificity for TCR-based therapy. However, they apply immunoprecipitation and mass spectrometry to test the specificity of an antibody, which is hardly a new concept. They have changed the title to specify that the method was applied to a TCR-like antibody. However, the term TCR-based therapy is still used in the short title, the abstract and several places in the manuscript. I think this is misleading. In their rebuttal letter, the authors claim that the method could potentially be used to test the specificity of TCRs with high avidity. I would argue that this is unlikely. One would need a TCR with a very high affinity and also have to produce it in a soluble form, which is known to be difficult. The main advantage of this method is the ability to identify off-targets for a cross-reactive TCR-like antibody. As the authors explain, a negative result is no guarantee that the TCR is mono-specific. One will therefore still have to perform killing assays. I am therefore not sure how this method will help in de-risking TCR-like antibodies.

Reviewer #2 (Remarks to the Author):

The authors have adequately addressed my comments, therefore I recommend publication of the revised manuscript.

Reviewer #3 (Remarks to the Author):

Some of the responses provided by the authors have led to some clarifications. However, now that the data tables have been provided, several concerns remain and some corrections need to be made.

In my opinion, the filtering data operated based on predictive HLA binding is quite risky for data interpretation. Peptides that are significantly enriched by the MAGE-A4 antibody, even though they are not expected to bind HLA-A*0.2:01, remain capable of being recognized by the antibody, especially when cells are lysed, apoptotic and thus release their cytoplasmic

contents.

Also, restricting the predictive HLA-peptide binding to the peptide length identified by MS (as done by the authors with NetMHCpan 4.1) can lead to some reactive peptides being missed. Indeed, although some short peptides are not significantly enriched, they would become so if the intensities of the peptides from the same group were summed.

The Supplementary Table 4 is filled with values inverted from those displayed in the volcano plots of Figure 5A (Lung).

The total absence of background generated by the MAGE-A4 antibody with the intestine tissue is quite unusual. Intensity values present in Supplementary Table 5 for the MAGE-A4 conditions compared to those with the control antibody are unusually low or totally absent. This results in a very unbalanced volcano plot, contrasting with the classical results obtained with this type of analysis and with those in Figure 2A performed with liver tissue.

The statistical test (Anova) used for the MS analysis is incorrect. There are only 2 conditions to compare. A two-sided Welch's t-test seems more appropriate. A minimum fold change would also be included to set the confidence threshold.

Unless I'm mistaken, the authors still display the killing assay (Figure 5B) as a percentage of control TCB. As requested previously, the authors should display raw values, including those of the control TCB condition to assess the background of cell killing resulting from these experiments (% of cells remaining compared to the starting number incubated for the assay).

Minor points:

Some syntaxes need to be corrected (ex: line 265).

V20230928

RESPONSES TO REVIEWER COMMENTS

(Authors responses in green, with citations from manuscript highlighted in italic)

REVIEWER COMMENTS

Reviewer #1 (Remarks to the Author):

A main concern with the original manuscript was that the authors describe the approach as a generic method to assess specificity for TCR-based therapy. However, they apply immuno-precipitation and mass spectrometry to test the specificity of an antibody, which is hardly a new concept. They have changed the title to specify that the method was applied to a TCR-like antibody. However, the term TCR-based therapy is still used in the short title, the abstract and several places in the manuscript. I think this is misleading.

We have corrected this oversight and have replaced all occurrences of TCR-based therapeutics with TCR-like antibodies to avoid any confusion.

In their rebuttal letter, the authors claim that the method could potentially be used to test the specificity of TCRs with high avidity. I would argue that this is unlikely. One would need a TCR with a very high affinity and also have to produce it in a soluble form, which is known to be difficult.

The main advantage of this method is the ability to identify off-targets for a cross-reactive TCR-like antibody. As the authors explain, a negative result is no guarantee that the TCR is mono-specific. One will therefore still have to perform killing assays. I am therefore not sure how this method will help in de-risking TCR-like antibodies.

We are stating in the manuscript that we believe that understanding physiological off-targets for molecules in therapeutic development will help to guide their maturation away from these interactions as we state in the discussion: " We anticipate that MS-based assays could guide drug alterations during the maturation cycles of the drug prior to entry into human trials, supporting a physiologically relevant safety profile."

With time, physiological interactomes will be integrated as training data for predictors, refining relevant off-target sequences for increased predictability of interfering specificities.

We also discuss the limitations of the assay, and highlight that MS may miss low abundance, and non-canonical sequences: *Discussion: "Importantly, state-of-the-art proteomics approaches are a discovery technology which suffer from incomplete spectral assignments, and therefore may miss peptide identifications due to low abundance and/or unknown sequence modifications."*

Reviewer #2 (Remarks to the Author):

The authors have adequately addressed my comments, therefore I recommend publication of the revised manuscript.

We thank the reviewer again for their comments and time.

Reviewer #3 (Remarks to the Author):

Some of the responses provided by the authors have led to some clarifications. However, now that the data tables have been provided, several concerns remain and some corrections need to be made.

In my opinion, the filtering data operated based on predictive HLA binding is quite risky for data interpretation. Peptides that are significantly enriched by the MAGE-A4 antibody, even though they are not

expected to bind HLA-A*02:01, remain capable of being recognized by the antibody, especially when cells are lysed, apoptotic and thus release their cytoplasmic contents.

During the immunopeptidomics enrichment, we and others have observed co-precipitation of other membrane molecules. Therefore, peptide enrichment always contains HLA-associated peptides that were not enriched through the antibody, but by co-purification. We have optimized our workflows for many years, but these co-precipitations are unavoidable. Therefore, filtering steps are necessary in order to extract specific information from our experiments. We, however, completely agree with the reviewer that there is the possibility that the TCR-like molecule interacts with other HLAs but think that such occurrences need to be interrogated in future research.

We have nevertheless tested two additional peptides that were not predicted to bind to HLA-A*02:01 but were the most significantly enriched with the highest enrichment factor with the MAGE-A4 antibody in both the primary hepatocyte immunoprecipitation, and the lung tissue dataset. As we now detail in supplementary figure 4 and 5, these peptides do not give rise to an activating signal in the Jurkat NFAT activation assay.

Also, restricting the predictive HLA-peptide binding to the peptide length identified by MS (as done by the authors with NetMHCpan 4.1) can lead to some reactive peptides being missed. Indeed, although some short peptides are not significantly enriched, they would become so if the intensities of the peptides from the same group were summed.

We assume that each peptide (which is analysed indeed using the summed intensity of all measured charge states), is a separate HLA ligand. We consider all lengths of peptides known to be able to bind to HLA class I molecules, which spans a length from 8 to 14 amino acids. Peptides outside of this range have not been shown to be able to bind to HLA to our knowledge.

Our methodology, like any mass spectrometry-based peptide identification approach, may miss identification of some peptide ligands, but what we are trying to emphasise is the discovery power of this novel approach, which was here for the first time applied to TCR-like antibodies. *Discussion: "Importantly, state-of-the-art proteomics approaches are a discovery technology which suffer from incomplete spectral assignments, and therefore may miss peptide identifications due to low abundance and/or unknown sequence modifications."*

The Supplementary Table 4 is filled with values inverted from those displayed in the volcano plots of Figure 5A (Lung).

We are extremely sorry for this error, which was thankfully spotted by the reviewer. We have corrected the inverse values accordingly. As a consequence, we now observe two peptides that show minimal but significant enrichment in the MAGE-A4 immunoprecipitation from lung tissues. We have therefore tested these peptides for reactivity in NFAT reporter assays and confirmed no reactivity. These data are now reported in Supplementary figure 5.

The total absence of background generated by the MAGE-A4 antibody with the intestine tissue is quite unusual. Intensity values present in Supplementary Table 5 for the MAGE-A4 conditions compared to those with the control antibody are unusually low or totally absent. This results in a very unbalanced volcano plot, contrasting with the classical results obtained with this type of analysis and with those in Figure 2A performed with liver tissue.

Yes, we agree that there is a high number of peptides that are significantly enriched by the SPVL3 antibody in this experiment. We believe that this is likely due to the fact that colon tissue may have low level expression of HLA-DQ, which is the specificity of the control antibody.

The statistical test (Anova) used for the MS analysis is incorrect. There are only 2 conditions to compare. A two-sided Welch's t-test seems more appropriate. A minimum fold change would also be included to set the confidence threshold.

The Anova as carried out here in an "between subjects design" considers unequal variances and is, with only two groups, equivalent to a Welch t-test. A variance stabilisation by arcsinh normalisation is how the software considers unequal variances, and different values from individual measurements of different peptide charge states are integrated, which is why there are slight differences to a simple Welch's test performed on the quantitative values provided for each peptide. We have added more detail to the method section to clarify this.

Unless I'm mistaken, the authors still display the killing assay (Figure 5B) as a percentage of control TCB. As requested previously, the authors should display raw values, including those of the control TCB condition to assess the background of cell killing resulting from these experiments (% of cells remaining compared to the starting number incubated for the assay).

We had in our previous response and amended manuscript added the raw value graphs for figure 5 as an additional supplementary figure, but had unfortunately forgotten to indicate this accurately in our response, and we apologise for the oversight. For clarity, we have now included the raw graphs in the main figure 5 and replaced the figures that were displaying the values as % of control TCB. We hope that this is now acceptable.

Minor points:

Some syntaxes need to be corrected (ex: line 265).

We corrected syntax throughout the manuscript including the above example (now in line 259ff).

REVIEWERS' COMMENTS

Reviewer #1 (Remarks to the Author):

The authors have addressed my most important points. Please correct this sentence "An inherent downside of TCR-based therapies including TCR-like antibodies" replace including with AND.

The results are convincing and support the conclusions.

Reviewer #3 (Remarks to the Author):

The authors have adequately addressed my comments.

REBUTTAL

Reviewer #1 (Remarks to the Author):

The authors have addressed my most important points. Please correct this sentence "An inherent downside of TCR-based therapies including TCR-like antibodies" replace including with AND.

DONE

The results are convincing and support the conclusions.

Reviewer #3 (Remarks to the Author):

The authors have adequately addressed my comments.

We thank all reviewers again for all their helpful comments and the time invested in reviewing our manuscript.